# Microtubule nucleation and γTuRC centrosome localization in interphase cells require ch-TOG

Aamir Ali[1], Chithran Vineethakumari [1], Cristina Lacasa[1] & Jens Lüders [1] ✉

Organization of microtubule arrays requires spatio-temporal regulation of the microtubule nucleator γ-tubulin ring complex (γTuRC) at microtubule organizing centers (MTOCs). MTOC-localized adapter proteins are thought to recruit and activate γTuRC, but the molecular underpinnings remain obscure. Here we show that at interphase centrosomes, rather than adapters, the microtubule polymerase ch-TOG (also named chTOG or CKAP5) ultimately controls γTuRC recruitment and activation. ch-TOG co-assembles with γTuRC to stimulate nucleation around centrioles. In the absence of ch-TOG, γTuRC fails to localize to these sites, but not the centriole lumen. However, whereas some ch-TOG is stably bound at subdistal appendages, it only transiently associates with PCM. ch-TOG's dynamic behavior requires its tubulin-binding TOG domains and a C-terminal region involved in localization. In addition, ch-TOG also promotes nucleation from the Golgi. Thus, at interphase centrosomes stimulation of nucleation and γTuRC attachment are mechanistically coupled through transient recruitment of ch-TOG, and ch-TOG's nucleation-promoting activity is not restricted to centrosomes.

Timely assembly and remodeling of microtubule arrays are crucial for cell proliferation, differentiation, and homeostasis. A disorganized microtubule network may impair motor-dependent transport, disrupt cytoplasmic organization, or interfere with faithful chromosome segregation during cell division[1–3]. Microtubules have an intrinsic polarity that is determined by the orientation of α-β-tubulin heterodimers within the microtubule lattice. In cells the so-called plus-end of microtubules is highly dynamic and free to explore the cytoplasm, whereas the minus-end is more stable and frequently anchored at MTOCs[4]. By nucleating and anchoring microtubules at MTOCs and by controlling the number and distribution of MTOCs, cells have powerful means to control overall microtubule network configuration[5,6].

An essential component required for the formation of nucleation sites at MTOCs is the microtubule nucleator γ-tubulin ring complex (γTuRC), a ~2 MDa protein complex composed of γ-tubulin and additional subunits. According to the template nucleation model, the circular, helical arrangement of γ-tubulin subunits in the γTuRC, which resembles the configuration of tubulins in the microtubule lattice, provides a template that facilitates α-β-tubulin assembly[7,8]. However, several recent cryo-electron microscopy studies have revealed a surprising asymmetry in the γTuRC structure that may not be compatible with efficient nucleation[9–12]. Consistent with this, purified γTuRC is a poor nucleator in vitro[9,10,13]. Thus, nucleation from MTOCs may require not only recruitment but also specific activation of γTuRC, possibly through a conformational change.

Indeed, studies in yeast and invertebrates have suggested that MTOC-resident adapter proteins such as Spc110 and Spc72 at the budding yeast spindle pole body (SPB), Mto1 at the fission yeast SPB and interphase MTOCs, or Cnn and SPD-5 at the centrosome in flies and worms, respectively, may function in both recruitment and activation of γTuRC[14–23]. These functions may be mediated by a shared, conserved CM1 motif that was shown to bind and activate γ-tubulin complexes[16,21,24–26]. However, recent in vitro reconstitution experiments using purified proteins have failed to observe activation of

[1]Institute for Research in Biomedicine (IRB Barcelona), The Barcelona Institute of Science and Technology, Barcelona 08028, Spain.
✉e-mail: jens.luders@irbbarcelona.org

human γTuRC by CM1[10]. Thus, whether CM1 directly activates γTuRC remains unclear.

The targeting factor most proximal to human γTuRC is NEDD1. NEDD1 broadly mediates targeting of γTuRC including to experimentally induced, ectopic sites[25,27], but does not activate γTuRC[25,28,29]. Instead, NEDD1-bound γTuRC was suggested to be involved in microtubule anchoring[25]. Curiously, while human homologs of CM1-containing adapters exist, their contributions to γTuRC recruitment and activation at MTOCs remain poorly defined.

Pericentrin (PCNT) and CDK5RAP2 are part of the PCM, a proteinaceous scaffold around the proximal part of centrioles. The PCM is generally considered to be the canonical nucleation site at the centrosome. Yet, both proteins are largely dispensable for centrosomal nucleation in interphase[30]. During mitosis they are required for PCM expansion and increased nucleation activity, but are not essential for centrosomes to organize spindle poles[31–33]. This is surprising since CDK5RAP2 and potentially also pericentrin contain a CM1 motif, and both interact with γTuRC[34–36].

CDK5RAP2 and its paralog PDE4DIP (also known as myomegalin) also localize to the Golgi, where they are involved in assembling the Golgi-associated MTOC, but again this contribution is not essential. The upstream scaffold protein AKAP9 is essential for the recruitment of γTuRC to the Golgi MTOC, but, in the absence of other Golgi-associated MTOC proteins, cannot promote γTuRC activation[37]. AKAP9 and PDE4DIP are also at the centrosome, but here they do not seem to affect γTuRC recruitment or nucleation activity[30]. Another scaffold protein, CEP192, is crucial for γTuRC centrosome recruitment both in interphase and mitosis, but there is no evidence that it can activate γTuRC[38–40]. Thus, in human cells γTuRC recruitment seems to depend on multiple adapters and targeting factors, potentially indicating the existence of distinct γTuRC sub-populations at MTOCs, but no single protein seems to account for robust activation of nucleation.

Early work in Xenopus laevis egg extract showed that reconstitution of nucleation activity to salt-stripped centrosomes required the presence of the microtubule polymerase XMAP215 in addition to γTuRC, although the underlying mechanism remained unclear[41]. XMAP215 family proteins contain arrays of TOG domains that display differential binding affinities towards soluble and lattice incorporated α-β-tubulin dimers[42–45]. While XMAP215 family members are mostly described as proteins that stimulate microtubule growth by promoting tubulin addition at microtubule plus-ends, opposing catastrophe factors such as kinesin-13 KIF2C/MCAK[46–49], they have also been implicated in microtubule nucleation by γ-tubulin complexes in invertebrates[50–53].

In vitro, purified Xenopus XMAP215 was shown to stimulate nucleation from microtubule seeds[54] and from purified, immobilized γTuRCs[13]. Its C-terminal part was found to directly bind γ-tubulin, whereas its tubulin-binding TOG domains were important for nucleation[13]. Similarly, in budding and fission yeast the XMAP215 family members Stu2 and Alp14, respectively, were shown to interact with γ-tubulin-complexes and stimulate nucleation at MTOCs through their TOG domains[50,51]. In contrast, affinity-purified human γTuRC was not associated with the XMAP215 family protein ch-TOG[24,55,56] and whether ch-TOG promotes γTuRC-dependent nucleation in vertebrate cells has not been investigated.

In addition to localizing to the PCM, γTuRC subunits have been found in association with the outer part of mother centriole-specific subdistal appendages and in the centriole lumen[57]. Whereas luminal γTuRC may not function as nucleator, but as a stabilizer that promotes centriole integrity[58], the role of γTuRC at subdistal appendages is still unclear. Subdistal appendages contribute to the centrosomal MTOC by anchoring microtubule minus-ends, but how and where these microtubules are generated is unknown. Thus, sub-distal appendage-bound γTuRC may be involved in microtubule nucleation, anchoring, or both.

Here, we have analyzed the roles of the XMAP215 family member ch-TOG at nucleation sites in human interphase cells. Using super resolution imaging we show that ch-TOG is crucial for microtubule nucleation from the centrosome by co-assembling with γTuRC and by stimulating nucleation. Our data further suggest that at the centrosome, subdistal appendages may function as a nucleation site, in addition to the canonical nucleation pathway in the PCM. Apart from centrosomes, ch-TOG also stimulates nucleation at the Golgi.

Consistent with ch-TOG being a transient interactor of γTuRC, ch-TOG localizes to nucleation sites during nucleation but disperses as microtubules elongate. An exception are subdistal appendages, where ch-TOG is more stably bound, suggesting additional functions. The requirement of ch-TOG for γTuRC attachment at interphase centrosomes indicates that rather than simply docking to an adapter protein as generally assumed, stable binding of γTuRC at the centrosomal MTOC depends on stimulation of nucleation by ch-TOG, suggesting that these two events are mechanistically linked.

## Results
### ch-TOG at interphase centrosomes is associated with outer subdistal appendages
To investigate whether ch-TOG may participate in microtubule nucleation from MTOCs in human cells we analyzed its previously described localization at centrosomes[41,48] by structured illumination microscopy (SIM) using antibodies against ch-TOG and acetylated tubulin to label centriole cylinders.

To our surprise we found that throughout most of interphase centrosome-bound ch-TOG was not associated with the PCM, but with the distal part of only one of the two centrioles, where its distribution resembled subdistal appendage localization. Indeed, ch-TOG partially colocalized with the subdistal appendage marker NIN (ninein) (Fig. 1a). This staining was observed with two different antibodies and demonstrated to be specific by RNAi-mediated depletion (Fig. 1a, b). The ch-TOG antibody #2 additionally labeled the distal end of both mother and daughter centrioles, but we did not investigate this localization further. Moreover, exogenously expressed, tagged ch-TOG showed a similar localization pattern (Fig. 1a).

In contrast to earlier cell cycle stages, in late S/G2 phase ch-TOG was detected on both mother centrioles where it decorated also more proximal centriole regions, in addition to its localization to subdistal appendages (Supplementary Fig. 1a). Knockdown of the inner subdistal appendage protein CEP128 eliminated ch-TOG staining, whereas knockdown of ch-TOG had no effect on subdistal appendage localization of CEP128 or ODF2, another inner subdistal appendage protein[59] (Fig. 1d, e). Thus, throughout most of the cell cycle centrosomal ch-TOG is associated with the outer region of subdistal appendages.

### A fraction of ch-TOG transiently localizes to centrosomes
To test if interphase centrosome localization of ch-TOG was independent of microtubules, as would be the case for an integral component of centrosomes, we depolymerized microtubules by treatment with nocodazole. Unexpectedly, microtubule depolymerization caused ch-TOG to strongly accumulate around both mother and daughter centriole cylinders, causing an increase in centrosomal ch-TOG signal. Again, this was observed with both antibodies (Fig. 2a, b; Supplementary Fig. 1a). Overall cellular ch-TOG levels were unchanged (Fig. 2d), indicating that its accumulation at centrosomes was due to specific recruitment. This behavior was also observed in a different human cell line, with exogenously expressed, tagged ch-TOG, and was independent of the cell cycle stage (Fig. 2a; Supplementary Fig. 1a, b, c). We confirmed the specificity of ch-TOG accumulation around the centrioles in the absence of the microtubules by RNAi (Supplementary Fig. 2a, b).

Accumulation of ch-TOG occurred mostly in the proximal centriole region, where it partially colocalized with γ-tubulin suggesting

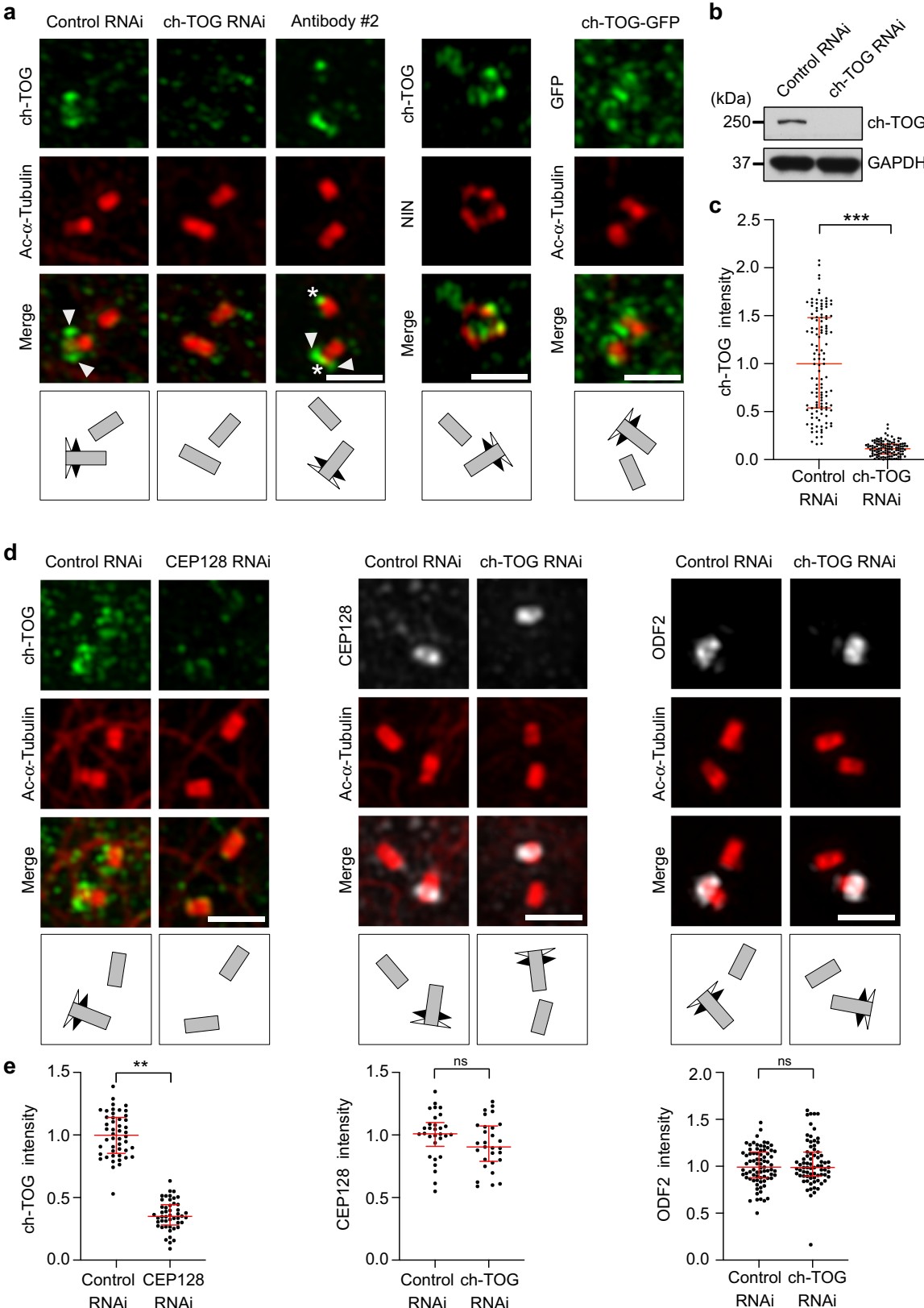

that it was present in the PCM (Fig. 2a, white arrows; Supplementary Fig. 2b). Careful analysis revealed that a small amount of γ-tubulin was also detectable at subdistal appendages, indicating that colocalization of ch-TOG and γTuRC occurred at both sites (Fig. 2a, white arrow heads; Supplementary Fig. 1d).

To dissect more precisely how microtubules affected ch-TOG centrosome localization, we quantified sub-centrosomal ch-TOG distribution in the presence and absence of microtubules. PCM-associated ch-TOG was only detected in the absence of microtubules and almost completely removed in their presence. Similarly, while

**Fig. 1 | ch-TOG localizes to the outer subdistal appendages. a** Maximum intensity projections of 3D-SIM images of centrosomal immunofluorescence staining of endogenous and recombinant ch-TOG (ch-TOG-GFP). U2OS cells transfected with control or ch-TOG siRNA #1, untreated U2OS cells, and U2OS cells stably expressing ch-TOG-GFP were stained with anti-ch-TOG antibody #1, anti-ch-TOG antibody #2 or anti-GFP antibody. Costaining was performed with antibodies against acetylated α-tubulin or NIN as indicated. Arrowheads point at subdistal appendage signals, asterisks indicate centriole distal end signals. **b** Lysate from U2OS cells transfected with control or ch-TOG siRNA #1 were analyzed by immunoblot against proteins indicated on the right. Detection of GAPDH served as loading control. Shown is one of two experiments with similar result. **c** Centrosomal ch-TOG staining intensity in cells after control or ch-TOG RNAi as in (**a**) normalized to the average of the control were quantified and plotted. Results are from $N = 3$ independent experiments, total number of cells analyzed: 60 in control and 58 in ch-TOG RNAi, ****$p = 0.0001$.

**d** U2OS cells transfected with control siRNA, CEP128 siRNA or ch-TOG siRNA were stained with antibodies against ch-TOG (antibody #1), CEP128 or ODF2 as indicated. Co-staining of acetylated α-tubulin was used to label centrioles. **e** Centriolar fluorescence intensities of the indicated proteins in cells as in (**d**) normalized to the average of the control were quantified and plotted. Results are from $N = 2$ independent experiments, total number of cells analyzed: 48 in control and CEP128 RNAi, **$p = 0.0028$ (ch-TOG staining); 30 in control and 29 in ch-TOG RNAi, $p = 0.4051$ (CEP128 staining); 77 in control and 74 in ch-TOG RNAi, $p = 0.2148$ (ODF2 staining). Statistical significance was determined by unpaired, two-tailed $t$ test with Welch's correction. Ns, not significant. The horizontal bars and whiskers indicate median and interquartile range, respectively, of the plotted data points. Illustrations indicate centriole orientations in the respective images. Scale bars, 1 μm. Source data are provided as a Source Data file.

subdistal appendages retained ch-TOG signal in the presence of microtubules, depolymerization further increased this signal (Fig. 2c). Thus, microtubules may promote dissociation of most of ch-TOG recruited to the PCM and of a fraction of ch-TOG recruited to subdistal appendages.

To address this more directly, we asked how ch-TOG, accumulated at centrosomes through microtubule depolymerization, would be affected if microtubules were allowed to regrow. Strikingly, as soon as 5 s after regrowth was initiated, the centrosome-associated ch-TOG signal started to disperse, and by 10 s it was not detectable anymore (Fig. 2e). Instead, ch-TOG puncta were detectable in a wider area around centrosomes. By 30 s ch-TOG staining remained detectable mostly at subdistal appendages, similar to ch-TOG in control cells in the presence of microtubules (Fig. 2e).

Costaining with α-tubulin showed that dispersal of centrosome-localized ch-TOG during regrowth occurred through association with microtubules, since the ch-TOG puncta frequently colocalized with the tips and lattices of outgrowing microtubules (Supplementary Fig. 1e). We conclude that a fraction of ch-TOG displays transient interaction with centrosomes, in which recruitment is followed by removal through association with outgrowing microtubules.

### ch-TOG and γTuRC are interdependent for centrosome localization

Since ch-TOG colocalized with γTuRC at subdistal appendages and transiently also in the PCM, we wondered whether ch-TOG affected γTuRC localization. We found that depletion of ch-TOG removed γ-tubulin from both sites, the subdistal appendages and the PCM, decreasing overall centrosomal γ-tubulin staining by ~50% (Fig. 3a, b). The remaining γ-tubulin signal was almost exclusively associated with the centriole lumen. Similar results were also obtained by ch-TOG knockdown with a different siRNA (Supplementary Fig. 2c, d, e).

Importantly, signals for the PCM marker PCNT (Fig. 3a, b) and the inner subdistal appendage protein CEP128 (Fig. 3c) were not reduced, and total levels of γ-tubulin were similar to controls (Supplementary Fig. 2a, c), indicating a specific reduction in centrosomal γ-tubulin. Moreover, loss of PCM-associated γ-tubulin after ch-TOG depletion was also observed in the absence of microtubules, conditions that in control cells resulted in ch-TOG PCM accumulation (Supplementary Fig. 2c). Centrosomal levels of NEDD1, which targets γTuRC to centrosomes[27,28], were similarly reduced, suggesting that loss of ch-TOG impaired NEDD1-dependent centrosomal attachment of γTuRC (Fig. 3a, b).

We then asked if ch-TOG functioned upstream of NEDD1 or was interdependent with it. To test this, we depleted NEDD1 and stained cells with ch-TOG and γ-tubulin antibodies. To our surprise, in NEDD1 depleted cells not only γTuRC but also ch-TOG failed to localize to centrosomes, both in the presence and absence of microtubules (Fig. 3d, e). Thus, ch-TOG and NEDD1-γTuRC are interdependent for

centrosome recruitment both at the subdistal appendages and at the PCM.

Apart from NEDD1, centrosome recruitment of γTuRC was proposed to involve the PCM adapter protein CDK5RAP2[35]. However, a more recent study conducted in RPE1 CDK5RAP2 KO cells found only a minor reduction in γ-tubulin levels at interphase centrosomes that lack CDK5RAP2[30]. Using a similar CDK5RAP2 KO line[37] we could confirm this result (Supplementary Fig. 3a, b). Importantly, in the CDK5RAP2 KO background knockdown of ch-TOG reduced centrosomal levels of γ-tubulin similar to the reduction observed in wild-type cells, indicating that the majority of γTuRCs on the outside of centrioles depend on ch-TOG rather than CDK5RAP2 (Supplementary Fig. 3a, b).

### Human ch-TOG and γTuRC are transient interactors

Consistent with our data that ch-TOG only transiently colocalized with γTuRC at the PCM, ch-TOG, and γTuRC have not been reported to form complexes in human cells (http://www.thebiogrid.org)[55,56,60]. Indeed, we were unable to detect an interaction by standard immunoprecipitation of endogenous ch-TOG or γTuRC subunits from human cell extract (Supplementary Fig. 3c, d, e). Exogenous expression and immunoprecipitation of GCP3 fused to the biotin ligase BirA also did not precipitate ch-TOG, whereas γ-tubulin was detected (Fig. 3f).

To test if ch-TOG and γTuRC interacted transiently, we used a proximity biotinylation assay. We incubated cells expressing FLAG-BirA-GCP3 with biotin to allow biotinylation of proteins in proximity of the fusion protein and performed streptavidin pull-down to isolate biotinylated proteins. In this case we could readily detect ch-TOG in pulldowns from FLAG-BirA-GCP3 expressing cells but not cells expressing FLAG-BirA alone (Fig. 3g). Similar results were obtained using a FLAG-GCP2-BirA fusion (Supplementary Fig. 3f). We conclude that in human cells γTuRC and ch-TOG may not form a stable complex but are transiently in close proximity.

### ch-TOG is required for nucleation at interphase centrosomes

To test the functional consequences of loss of γTuRC from PCM and subdistal appendages in ch-TOG knockdown cells we performed microtubule regrowth assays. Microtubules were depolymerized by incubation of cells on ice and allowed to regrow by immersion of cells in medium maintained at 37 °C. After 5 s of regrowth, small centrosomal microtubule asters had formed in control cells (Fig. 4a).

Maximum intensity projection of 3D-SIM imaging of these asters revealed that microtubules were not evenly distributed around centrioles. Instead, a majority of microtubules was associated with one of the centrioles, which we identified as mother centriole by ODF2 staining. Microtubules seemed to originate from multiple sites around centrioles including ODF2-positive sites. In ch-TOG-depleted cells microtubule nucleation was strongly inhibited and only very few

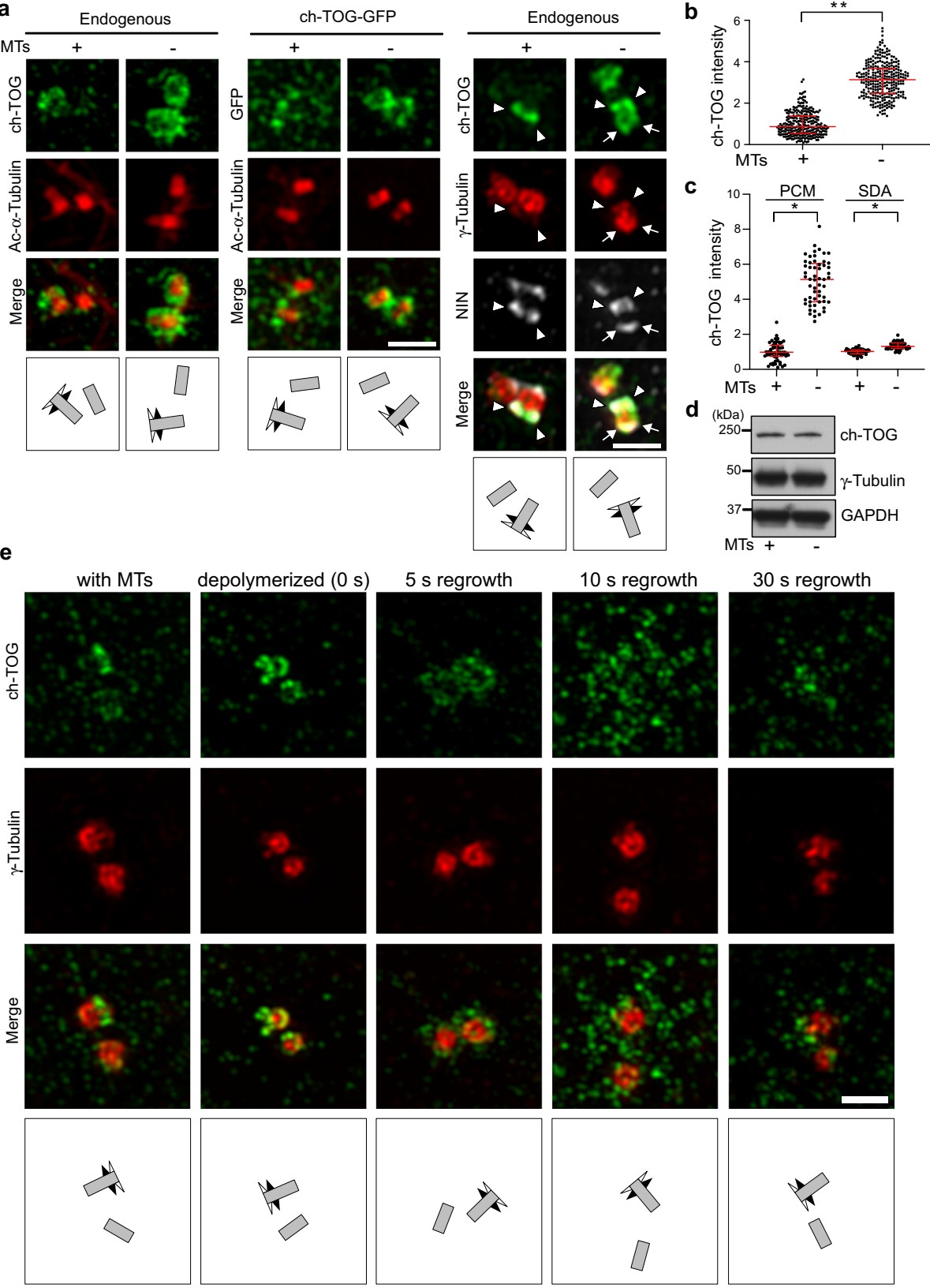

centrosomal microtubules could be detected, mostly in association with mother centrioles (Fig. 4a, b). Here, in the absence of bulk nucleation, a subset of individual microtubules were more clearly seen to grow from the subdistal appendage region marked by ODF2 (Fig. 4a, yellow arrowheads).

To test whether nucleation may occur from subdistal appendages, we performed an additional regrowth experiment in untreated cells. After regrowth for 2 s, we fixed cells and co-stained microtubules with γ-tubulin or the outer subdistal appendage protein NIN. We observed that NIN and γ-tubulin signals at subdistal appendages coincided with

**Fig. 2 | ch-TOG transiently localizes to the PCM. a** Maximum intensity projections of 3D-SIM images of centrosomes stained for endogenous or recombinant ch-TOG (ch-TOG-GFP), in the presence or absence of microtubules. Cells were costained with antibodies against acetylated α-tubulin or antibodies against γ-tubulin and NIN as indicated. **b** Total centriolar intensities of endogenous ch-TOG staining in cells with or without microtubules as in (**a**) were quantified, normalized to the average intensity in cells with microtubules, and plotted. $N = 3$ independent experiments, total number of cells analyzed per condition: 141 (MTs + ) and 144 (MTs -), respectively. **$p = 0.0018$. The horizontal bars and whiskers indicate median and interquartile range, respectively, of the plotted data points. Statistical significance was determined by unpaired, two-tailed t test with Welch's correction. **c** Intensities of endogenous ch-TOG staining at the PCM and at subdistal appendages in cells with or without microtubules as in (**a**) were quantified, normalized to the average intensities in cells with microtubules, and plotted. $N = 3$ independent experiments. Total number of cells analyzed for PCM staining: 55 (MTs+) and 56 (MTs−). *$p = 0.0368$. Total number of cells analyzed for subdistal appendage staining: 55(MTs+) and 56 (MTs−). *$p = 0.0142$. The horizontal bars and whiskers indicate median and interquartile range, respectively, of the plotted data points. Statistical significance was determined by unpaired, two-tailed t test with Welch's correction. **d** Cell lysates prepared from U2OS cells with and without microtubules were analyzed by immunoblot using antibodies against the indicated proteins. Observed in two independent experiments. **e** Microtubules in U2OS cells were depolymerized, allowed to re-grow for the indicated time points, fixed and stained with antibodies against the indicated proteins. Illustrations indicate centriole orientations in the respective images. Performed twice with similar result. Scale bar, 1 μm. Source data are provided as a Source Data file.

the ends of regrown microtubules, consistent with nucleation from these sites (Fig. 4c, d, yellow arrowheads).

### ch-TOG uses distinct domains for localization and stimulation of nucleation

To gain mechanistic insight into how ch-TOG targets to distinct centrosomal sites and co-assembles with γTuRC, we performed rescue experiments. We generated stable cell lines expressing RNAi-resistant, GFP-tagged full length ch-TOG and deletion mutants (Fig. 5a, b; Supplementary Fig. 4a, b). After transfection of control or ch-TOG siRNA we determined centrosome localization of the rescue constructs and of endogenous γ-tubulin.

Full length ch-TOG (ch-TOG) was predominantly associated with subdistal appendages, similar to endogenous ch-TOG, and in knockdown cells rescued loss of γ-tubulin staining both at the subdistal appendages and the PCM (Fig. 5c, d). Constructs comprising the first two or all five TOG domains, but lacking the C-terminal domain (ch-TOG-12 and ch-TOG-12345, respectively), did not localize to centrosomes, indicating that the C-terminal region is involved in centrosome targeting[48] (Fig. 5c; Supplementary Fig. 4c).

Indeed, the C-terminal fragment alone (ch-TOG-C) targeted to subdistal appendages independently of endogenous ch-TOG. However, it was unable to rescue γ-tubulin recruitment, which was only detectable in the centriole lumen (Fig. 5c, d). In the case of *Xenopus* XMAP215, a slightly larger fragment comprising the C-terminal region and the preceding TOG domain was shown to interact with purified γ-tubulin in vitro[13]. In human cells the corresponding fragment (ch-TOG-5C) behaved similar to ch-TOG-C, localizing to subdistal appendages and failing to recruit γ-tubulin (Supplementary Fig. 4c, d).

Including two additional N-terminal TOG domains (ch-TOG-345C) still did not rescue γTuRC recruitment, but, interestingly, led to some PCM localization of this construct in the presence of endogenous ch-TOG. This was not observed in cells depleted of endogenous ch-TOG, where the protein was largely displaced from centrosomes (Fig. 5c, d). Thus, compared to full-length ch-TOG this fragment appears to bind more stably to the PCM, as long as γTuRC is recruited there through endogenous ch-TOG, but cannot itself support stable PCM integration of γTuRC in the absence of endogenous ch-TOG.

Consistent with the inability of ch-TOG deletion mutants to rescue γTuRC centrosome recruitment, centrosomal nucleation activity was strongly reduced in all cases, and was rescued only by expression of full length ch-TOG (Fig. 5e; Supplementary Fig. 4e, f, g).

In summary, the data show that ch-TOG localization to subdistal appendages and PCM requires its C-terminal region. N-terminal TOG domains are additionally needed to stimulate nucleation, resulting in transient PCM interaction and incorporation of γTuRC at this site.

### ch-TOG functions at non-centrosomal nucleation sites

We wondered whether the transient recruitment and microtubule-dependent dissociation of ch-TOG only occurred at centrosomes or was a more general mechanism to control nucleation. We first asked if such behavior could also be observed at centrosomes during mitosis, a stage at which the centrosomal MTOC is reorganized through the expansion of the PCM[61] and the partial disassembly of subdistal appendages[62]. ch-TOG was readily detectable at the mitotic centrosomes that formed the spindle poles. Upon microtubule depolymerization the centrosomal ch-TOG signal appeared to be more confined to the PCM region and significantly increased, suggesting that at least a fraction of ch-TOG also displayed transient PCM interaction (Fig. 6a, b). However, consistent with a previous report[63], ch-TOG depletion did not significantly reduce γ-tubulin signals at mitotic centrosomes, suggesting that during mitosis other robust recruitment mechanisms exist (Supplementary Fig. 5).

We then analyzed non-centrosomal nucleation in interphase RPE1 cells, most of which occurs from the surface of Golgi membranes[64]. We depolymerized microtubules by treatment with nocodazole, washed out the drug on ice, and then incubated in warm medium to allow microtubule nucleation. After 10 s of regrowth we could detect multiple short non-centrosomal microtubules that had both γ-tubulin and ch-TOG signals at one end. After 30 s of regrowth, only γ-tubulin remained detectable at the ends of microtubules, where ch-TOG signals were rarely observed. This showed that transient recruitment of ch-TOG occurred not only at centrosomes, but also at non-centrosomal nucleation sites.

### ch-TOG promotes nucleation from Golgi membranes

We sought to test whether ch-TOG was not only present, but also promoted nucleation at non-centrosomal sites. For this we analyzed the Golgi-associated MTOC in RPE1 cells after nocodazole treatment to completely depolymerize microtubules and disperse the Golgi, which allows for visualization of nucleation from individual stacks[65].

After triple labeling with antibodies against α-tubulin, ch-TOG and the cis-Golgi marker GM130 we found that not all Golgi membranes were associated with microtubules, but that nucleation occurred in clusters. This was consistent with the presence of nucleation 'hotspots' that are heterogeneously distributed across the Golgi[65]. ch-TOG signals were occasionally observed at short microtubules in these clusters, but could not be assigned to a specific microtubule end and were also abundant in the surrounding cytoplasm (Supplementary Fig. 6).

Instead, we tested if ch-TOG was required for nucleation from the Golgi by performing regrowth experiments after ch-TOG knockdown. Quantification of the number of microtubules that had formed at the Golgi revealed significantly reduced activity in ch-TOG-depleted cells compared to controls (Fig. 7a, b). Together the data show that ch-TOG promotes γTuRC-dependent nucleation not only at centrosomes, but also at non-centrosomal nucleation sites associated with the Golgi.

### Discussion

Recruitment and activation of the microtubule nucleator γTuRC are fundamental to the function of MTOCs. In this study, we show that at

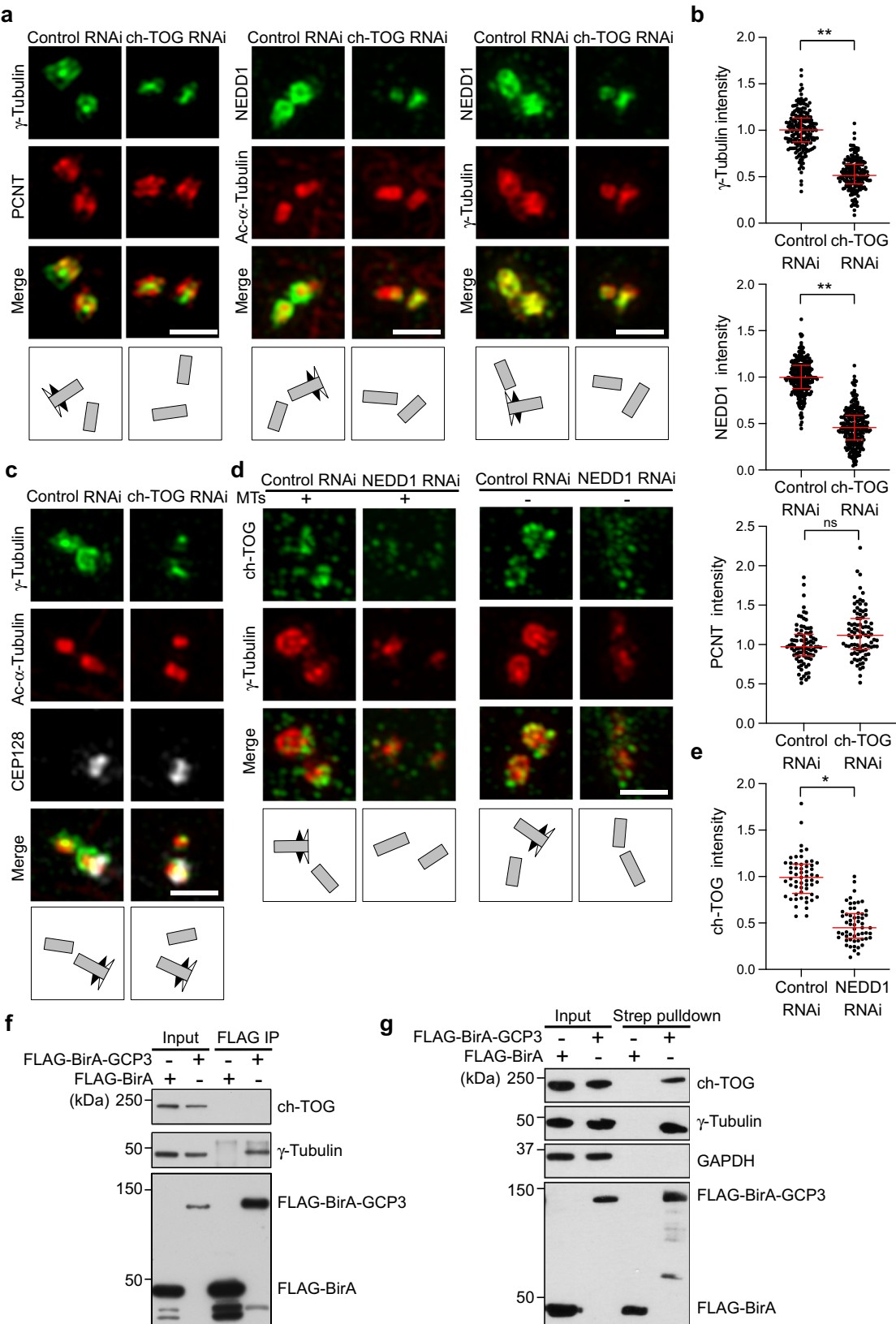

interphase centrosomes in human cells the XMAP215 family member ch-TOG is crucial for both of these functions. Paradoxically, we found that ch-TOG neither forms a stable complex with γTuRC nor does it localize stably to the PCM, the canonical centrosomal nucleation site. How, then, does ch-TOG support γTuRC centrosome recruitment and nucleation?

While we did not detect interaction between ch-TOG and γTuRC by immunoprecipitation, it could readily be observed by biotin proximity labeling. Moreover, in a previous study, XMAP215 was found to co-immunoprecipitate with γTuRC subunits in *Xenopus* egg extract[13]. Thus, we conclude that γTuRC and ch-TOG do interact, but in

**Fig. 3 | ch-TOG promotes incorporation of γTuRC at PCM and subsidal appendages. a** U2OS cells transfected with control or ch-TOG siRNA #1 were fixed and stained with antibodies against the indicated proteins. **b** Intensities of centriolar γ-tubulin, NEDD1, and PCNT staining in cells as in (a) were quantified, normalized to the average of the respective controls, and plotted. γ-tubulin staining: *N* = 3 independent experiments; total number of cells analyzed, 75 (Control RNAi) and 69 (ch-TOG RNAi); ***p* = 0.0004. NEDD1 staining: *N* = 3 independent experiments; total number of cells analyzed, 312 (Control RNAi) and 327 (ch-TOG RNAi); ***p* = 0.0042. PCNT staining: *N* = 2 independent experiments; total number of cells analyzed, 151 (Control RNAi) and 129 (ch-TOG RNAi); *p* = 0.3865 (ns, not significant). The horizontal bars and whiskers indicate median and interquartile range, respectively, of the plotted data points. Statistical significance was determined by unpaired, two-tailed *t* test with Welch's correction without multiple comparison. **c** U2OS cells transfected with control or ch-TOG siRNA #1 were co-stained with antibodies against the indicated proteins. Performed twice with similar result.

**d** U2OS cells treated with control or NEDD1 siRNA, with or without microtubules, were co-stained with antibodies against ch-TOG and γ-tubulin. **e** Centriolar ch-TOG intensities were quantified as in (d), normalized to the average of the intensities of the control, and plotted. *N* = 2 experiments, total number of cells analyzed, 30 per condition, ***p* = 0.0015. The horizontal bars and whiskers indicate median and interquartile range, respectively, of the plotted data points. Statistical significance was determined by unpaired, two-tailed *t* test with Welch's correction. Illustrations indicate centriole orientations in the respective images. Scale bars, 1 μm.
**f, g** Lysates from HEK293T cells transiently expressing Flag-BirA or Flag-BirA-GCP3 and grown in the presence of biotin for 24 h were subjected to affinity pulldowns using anti-FLAG antibody and streptavidin-coupled beads. The pulldown precipitates were subjected to immunoblot and probed with anti-ch-TOG, anti-γ-Tubulin, anti-FLAG, and anti-GAPDH antibodies. Detection of GAPDH was used as control. The results were replicated in two independent experiments. Source data are provided as a Source Data file.

human somatic cells the interaction may be transient or relatively unstable.

In microtubule nucleation assays we further found that ch-TOG recruitment to the PCM and colocalization with γTuRC occurred transiently and was detected only upon microtubule depolymerization, followed by dispersion of ch-TOG during microtubule outgrowth. This mechanism was unexpected, since current models assume a more conventional mechanism involving stable docking of γTuRC to an MTOC-bound adapter, followed by the activation of γTuRC nucleation activity[2,5,6,66–68]. Indeed, in vertebrates several MTOC-localized scaffold proteins have been identified and suggested to provide docking sites for γTuRC and stimulate its nucleation activity, but so far, no single factor was found to be crucial.

The molecular basis of this regulation is best understood in yeast. Adapters such as budding yeast Spc72 and Spc110 or fission yeast Mto1 interact with relatively inactive γ-tubulin complexes through their CM1 domain, promoting their recruitment and arrangement into active nucleation templates[15–17,20,21,69]. Nucleation is further promoted by XMAP215 family members such as budding yeast Stu2 or fission yeast Alp14[50–52].

In budding yeast, direct binding of the Spc72 adapter via its CM1 motif to the γ-tubulin small complex (γTuSC) is enhanced by additional interaction with a C-terminal region of Stu2, promoting γTuSC oligomerization into a nucleation template. The action of two N-terminal TOG domains in Stu2 then allows for optimal nucleation activity[51].

The situation in fission yeast is similar. Here the CM1-containing Mto1 adapter recruits γ-tubulin complexes to cytoplasmic MTOCs including the nuclear envelope. Alp14 then binds to stimulate nucleation[50,52]. Interestingly, even though Alp14 was shown to form stable complexes with Mto1 and γ-tubulin complexes, in cells its recruitment to nucleation sites may be transient[50], similar to what we have observed for ch-TOG. However, contrary to our findings, fission yeast γ-tubulin complexes can be recruited to nucleation sites independently of Alp14[52], suggesting that the two events, γ-tubulin complex recruitment and nucleation, are not mechanistically coupled in this organism.

In contrast, our results indicate that at human interphase centrosomes ch-TOG is crucial for stable incorporation of γTuRC through a mechanism that involves stimulation of nucleation. Nucleation of a microtubule, stimulated by ch-TOG, may induce a conformational change in γTuRC that promotes its binding to MTOC-bound adapters. Consistent with this model are our results from rescue experiments. The C-terminal region that mediates ch-TOG centrosome targeting[48] and the N-terminal TOG1 and TOG2 domains that stimulate nucleation[13] were both required to rescue depletion of endogenous ch-TOG. Thus, centrosome recruitment of γTuRC and stimulation of nucleation activity are provided by ch-TOG as non-separable functions, suggesting that they are coupled. This is further supported by

our finding that ch-TOG interphase centrosome localization, in turn, also depends on γTuRC.

Of note, some studies have implicated XMAP215 family members in microtubule nucleation independently of γTuRC[70–72]. We believe that this is not the case at unperturbed interphase centrosomes, since we found that ch-TOG and γTuRC depend on each other for centrosome localization. However, our data do not exclude that ch-TOG may, depending on the cell type, cell cycle stage or nucleation site, also provide γTuRC- and thus template-independent nucleation activity.

Since ch-TOG was found to only transiently associate with PCM, what is the adaptor that allows stable docking of γTuRC upon ch-TOG-stimulated nucleation? Bonafide PCM scaffold proteins such as AKAP9, pericentrin or CDK5RAP2, previously described as γTuRC recruitment factors[34–36], were recently shown in knockout cell line models to only marginally affect γTuRC recruitment and nucleation at interphase centrosomes[30]. The authors observed a much stronger contribution for CEP192, which we recently showed to recruit γTuRC not only to the PCM but along the entire centriole surface[58]. Thus, CEP192 may be the most important centrosomal adaptor for γTuRC recruitment.

We observed that the γTuRC-associated targeting factor NEDD1 was also lost from the centrosome in ch-TOG-depleted cells, suggesting that ch-TOG promotes γTuRC centrosome recruitment at the level of NEDD1's binding to the centrosomal scaffold rather than its interaction with γTuRC. In agreement with this possibility, CEP192 and NEDD1 were found to be in close proximity by biotin proximity labeling[73].

In contrast to ch-TOG's transient localization to the PCM, it was more stably associated with subsidal appendages, where it was readily detectable in steady-state conditions. ch-TOG was present at the outer subsidal appendage region, partially colocalizing with NIN. In agreement with this localization and with the hierarchical organization of subsidal appendage proteins, depletion of the inner subsidal appendage component CEP128 removed ch-TOG from these structures, whereas these proteins were unaffected by the absence of ch-TOG. Subsidal appendage localization was also observed for a C-terminal fragment of ch-TOG lacking the TOG domains. Based on these observations, it is tempting to speculate that ch-TOG may also be involved in microtubule anchoring, possibly using its C-terminal end for subsidal appendage association and one or more TOG domains for binding to microtubules.

By performing microtubule nucleation assays in cells, we made another important observation. Apart from the PCM, microtubules regrew also from more distal regions and this activity also required ch-TOG. Consistently, we found that γ-tubulin colocalized with ch-TOG at subsidal appendages, and that a fraction of subsidal appendage-associated ch-TOG displayed transient localization at this site similar to its behavior at the PCM.

Whereas subsidal appendages are commonly described as structures that anchor microtubule minus-ends, our results suggest

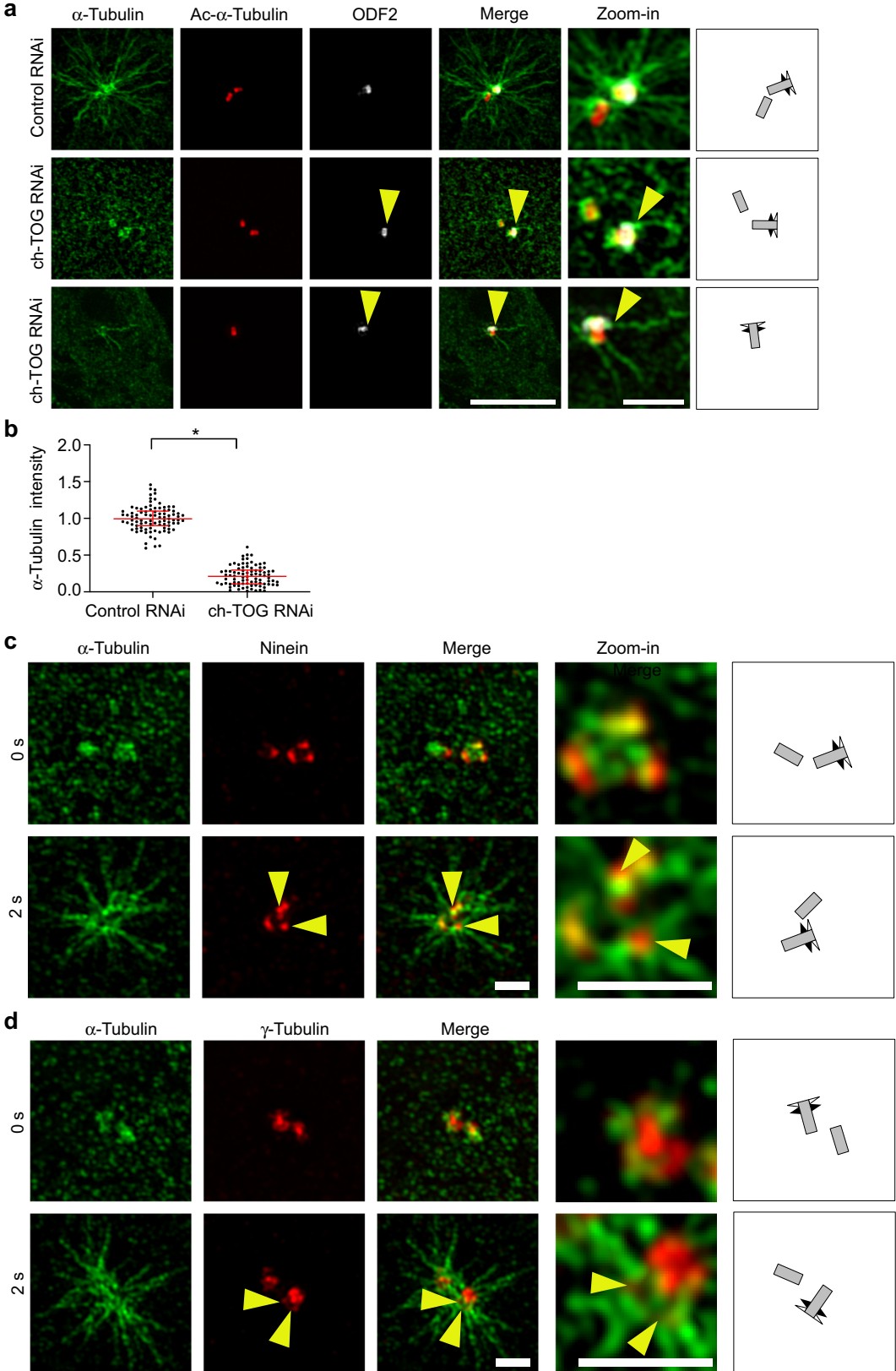

that they may also serve as nucleation site. This finding may resolve a long-standing conundrum: microtubule nucleation and anchoring seem to occur at spatially separated sites, the PCM, and the subdistal appendages, respectively. This raises the question of how minus-ends of newly nucleated microtubules are transferred to the anchoring site. We propose that subdistal appendages may perform both actions, nucleation and anchoring. Whether PCM-nucleated microtubules are subsequently transferred to the subdistal appendages[74,75], released[76,77], or anchored at their site of origin in a subdistal appendage-independent manner remains to be tested in future work.

Other important questions are whether ch-TOG cooperates with γTuRC also during mitosis, where multiple nucleation pathways

**Fig. 4 | ch-TOG is required for nucleation at interphase centrosomes. a** U2OS cells were transfected with control or ch-TOG siRNA. Microtubules were depolymerized by cold treatment for 30 min and then allowed to regrow for 5 s at 37 °C before fixation and staining with antibodies against the indicated proteins. Scale bar, 4 μm. Magnifications in the last column show centrosome regions. Performed twice with similar result. Scale bar, 1 μm. **b** α-Tubulin intensities around centrosomes were quantified, normalized to the average of the intensities of the control and plotted. *N* = 2 experiments, total number of cells analyzed: 91 (Control RNAi) and 86 (ch-TOG RNAi). *p = 0.0177. The horizontal bars and whiskers indicate median and interquartile range, respectively, of the plotted data points. Statistical significance was determined by unpaired, two-tailed *t* test with Welch's correction. **c, d** Microtubules were depolymerized by cold treatment for 30 min before allowing regrowth for 2 s by incubation at 37 °C. Cells were fixed and stained as indicated. Illustrations show centriole orientations in the respective images. Magnifications in the last column show centrosome regions. Regrowth from subdistal appendage area was observed in two independent experiments. Scale bars, 1 μm. Source data are provided as a Source Data file.

contribute to spindle assembly[1], and in different cell types, where non-centrosomal MTOCs such as the Golgi or the nuclear envelope contribute to microtubule organization[2,5].

We observed that upon microtubule depolymerization levels of ch-TOG at mitotic centrosomes were increased and more confined to the PCM, suggesting that some ch-TOG may transiently interact with mitotic centrosomes as in interphase. However, earlier work found that contrary to our results in interphase cells, γ-tubulin levels at mitotic centrosomes of human somatic cells were not altered after depletion of ch-TOG[63], a result that we have confirmed. Thus, mitotic centrosomes seem to employ additional, ch-TOG-independent γTuRC recruitment mechanisms.

During spindle assembly ch-TOG may also cooperate with γTuRC in chromatin- or augmin-mediated nucleation. Supporting this notion, ch-TOG, and TPX2, a factor of the chromatin-mediated nucleation pathway, were shown to function as minimal nucleation module in vitro[70]. As discussed by the authors, in cells this module likely involves γTuRC as nucleation template.

In *Drosophila* S2 cells, the augmin subunit Dgt6 was shown to interact with both γ-tubulin and XMAP215/Msps, and all proteins were shown to participate in chromatin-mediated nucleation[78]. Thus, while spindle defects after inhibition of ch-TOG or its homologs have generally been attributed to impaired microtubule stabilization and/or plus end growth, we propose that impaired nucleation may cause or at least contribute to some of the mitotic phenotypes that result from ch-TOG deficiency. However, additional work is required to establish ch-TOG's precise distribution and localization dependencies during mitosis.

Regarding non-centrosomal nucleation in interphase, microtubule regrowth assays revealed ch-TOG foci at one end of most short microtubules that formed in the cytoplasm away from centrosomes. γ-Tubulin was also found at microtubule ends suggesting that ch-TOG and γTuRC were present in nucleation complexes associated with microtubule minus-ends.

Similar to ch-TOG localization at the PCM, localization of ch-TOG at ends of non-centrosomal microtubules was not observed at later time points, whereas γ-tubulin could still be detected. Thus, transient recruitment of ch-TOG to γTuRC-dependent nucleation sites does not seem to be restricted to centrosomes. Indeed, γ-tubulin and XMAP215/Msps were shown to function together in the initial fast regrowth after depolymerization of the non-centrosomal interphase microtubule network in cultured *Drosophila* S2 cells[53]. Consistently, we found that nucleation from the non-centrosomal MTOC at the Golgi in RPE1 cells also involved ch-TOG.

Taken together, our results establish human ch-TOG as an important co-factor of γTuRC that, through transient interaction, links stimulation of microtubule nucleation to incorporation of γTuRC at interphase centrosomes. Moreover, the nucleation-promoting activity of ch-TOG also occurs at the Golgi MTOC, suggesting that it may be a more broadly used mechanism.

## Methods
### Cell culture
U2OS (human osteosarcoma, female) were cultured and maintained in DMEM containing 4.5 g/L of D-glucose, L-gutamine and pyruvate.

hTERT RPE-1 (human retinal pigment epithelial-1, female) were cultured and maintained in DMEM/F12 (1:1) containing L-Glutamine and 15 mM HEPES. DMEM and DMEM/F12 (1:1) were supplemented with 10% fetal bovine serum (FBS) and penicillin/streptomycin to culture U2OS and RPE-1 cells respectively at 37 °C. RPE1 CDK5RAP2 KO cells were described previously[37].

### Cloning and site-directed mutagenesis
A plasmid encoding full length ch-TOG was a gift from Stephen Royle (Addgene plasmid # 69112; http://n2t.net/addgene:69112; RRID:Addgene_69112)[79]. Site directed mutagenesis was used to generate full length, siRNA resistant ch-TOG (ch-TOG). Truncated cDNA constructs encoding ch-TOG-12, ch-TOG-15, ch-TOG-C, ch-TOG-345C, ch-TOG-5C were PCR amplified using Phusion polymerase, Dpn1 treated and subsequently ligated with T4 DNA ligase. All the cDNA mutations and truncations were verified by sequencing. FLAG-tagged BirA fusions of GCP2 and GCP3 were cloned in vector backbones as previously described[80].

### Generation of stable cell lines
U2OS cells stably expressing ch-TOG, ch-TOG-12, ch-TOG-15, ch-TOG-C, ch-TOG-345C, ch-TOG-5C were generated by plasmid transfection using Lipofectamine 2000 (Invitrogen). The drug-resistant transfected cells were selected and maintained using 200 μg/ml of geneticin. Following selection, cells positive for GFP fluorescence were sorted using fluorescence-activated cell sorting (FACS) and the recombinant protein expression was verified by immunoblot using anti-GFP (1:2000, Torrey Pines Biolabs) and anti-ch-TOG #2 (1:250, Santa Cruz Biotechnology sc-374394) antibodies.

### siRNA transfections
Human ch-TOG [siRNA #1: (5′-GAGCCCAGAGUGGUCCAAAdTdT-3′)[81]; siRNA #2 (5′-ACAUGCUCCACAGCAAACUCUdTdT-3′)[63], NEDD1 (5′-GCAGACAUGUGUCAAUUUGdTdT-3′), CEP128 (5′-GGAGCUAUCUCGAAGGUUAdTdT-3′), and control Luciferase (5′-UCGAAGUAUUCCGCGUACG-3′) siRNA transfections were performed as follows: 100 nM of siRNA and 2 μl of Lipofectamine RNAiMAX reagent were independently mixed in 250 μl of OPTI-MEM medium. After incubation and combining, siRNA-Lipofectamine RNAiMAX mixture was added to cells in 1.5 ml of medium without serum and antibiotic. The transfection medium was replaced with DMEM containing 4.5 g/L of D-glucose, L-glutamine and pyruvate after 6 h of transfection. The cells were harvested 72 h after the siRNA transfection unless otherwise specifically mentioned in the figure legends. The harvested cells were subjected to analysis by either immunoblotting or immunofluorescence.

### Microtubule depolymerization and regrowth assays
For studying centrosomal ch-TOG localization in the absence of microtubules, microtubules were depolymerized by treatment with nocodazole. Regrowth assays were performed as detailed previously[82]. To determine nucleation activity at centrosomes in U2OS cells, microtubules were depolymerized by incubation of cells grown on coverslips in culture dishes on an ice-water bath for 30 min. To visualize nucleation from Golgi membranes, RPE1 cells grown on

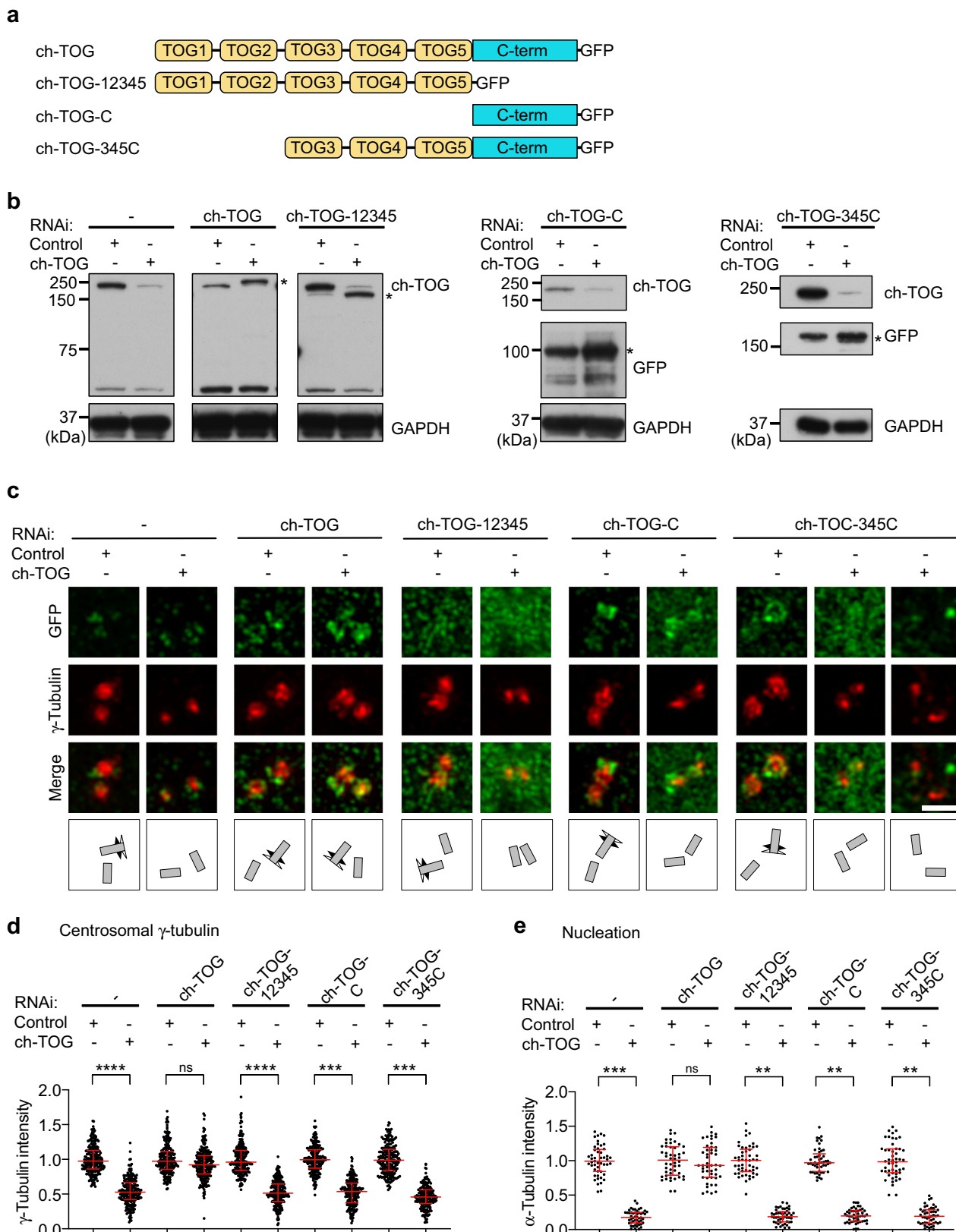

coverslips were treated with nocodazole (1.6 µg/ml) for 2 h. Nocoda-zole was washed out by 3 washes with chilled PBS and incubation on ice for 5 min at each wash. After the final wash, PBS was replaced with pre-chilled media and the cells were incubated for 30 min on an ice-water bath. The ice-water bath with culture dishes was placed next to pre-warmed medium kept in a 37 °C water bath and a dish with methanol

(pre-chilled at −20 °C). For the "0 s" time point, coverslips were removed with forceps from the dish with ice-cold medium and immediately immersed in pre-chilled methanol. For regrowth, cover-slips from the ice-cold medium were immersed in pre-warmed 37 °C medium, hand-held there for the indicated time, before immediately transferring them to methanol, pre-chilled at −20 °C, for fixation. This

**Fig. 5 | Centrosomal nucleation requires ch-TOG TOG domains and C-terminus.**
**a** Schematic representation of the domain structure of recombinant ch-TOG and truncation mutants carrying a C-terminal GFP tag. TOG domains colored in yellow, C-terminal domain in cyan. **b** U2OS wild type cells or cells stably expressing recombinant GFP-tagged ch-TOG constructs as in (**a**) were transfected with control or ch-TOG siRNA #1. Cell lysates were probed by western blotting for the indicated proteins. The asterisks mark the positions of recombinant proteins. Note that recombinant protein expression levels were always slightly increased in cells depleted of endogenous ch-TOG. **c** Cells as in (**b**) were analyzed by 3D-SIM for centrosomal localization of GFP-tagged ch-TOG constructs and of γ-tubulin. Illustrations indicate centriole orientations in the respective images. Scale bar, 1 μm. **d** Centriolar γ-tubulin intensities were quantified in cells as in (**c**), normalized to the average of the intensities of the control, and plotted. $N = 4$ experiments, total number of cells analyzed for control and ch-TOG RNAi, respectively: 116 and 116 (-); 120 and 117 (ch-TOG), 122 and 120 (ch-TOG-12345), 112 and 113 (ch-TOG-C), 113 and 95 (ch-TOG-345C). ***$p = 0.0002$ (-), $p = 0.4899$, not significant (ch-TOG),

****$p < 0.0001$ (ch-TOG-12345), ****$p < 0.0001$ (ch-TOG-C), ***$p = 0.0005$ (ch-TOG-345C). The horizontal bars and whiskers indicate median and interquartile range, respectively, of the plotted data points. Statistical significance was determined by unpaired, two-tailed $t$ test with Welch's correction. **e** Cells as in (**c**) were subjected to microtubule depolymerization by incubation on ice at 4 °C for 30 min. Following microtubule regrowth for 5 s at 37 °C, α-tubulin intensities of microtubule asters around centrosomes were quantified, normalized to the average of the intensities of the control, and plotted. $N = 2$ experiments, total number of cells analyzed for control and ch-TOG RNAi, respectively: 49 and 50 (-); 48 and 47 (ch-TOG), 50 and 48 (ch-TOG-12345), 49 and 50 (ch-TOG-C), 50 and 49 (ch-TOG-345C). **$p = 0.0084$ (-), $p = 0.3937$, not significant (ch-TOG), *$p = 0.0169$ (ch-TOG-12345), **$p = 0.0062$ (ch-TOG-C), *$p = 0.0136$ (ch-TOG-345C). The horizontal bars and whiskers indicate median and interquartile range, respectively, of the plotted data points. Statistical significance was determined by unpaired, two-tailed $t$ test with Welch's correction. Source data are provided as a Source Data file.

setup ensured precise timings and that the time between immersions in the different conditions was minimal (-1 s).

For quantification of microtubule regrowth from the Golgi, cells grown on coverslips were subjected to microtubule depolymerization by incubation on an ice-water bath for 30 min. Microtubules were allowed to regrow by immersing the coverslips for different time points in medium pre-heated to 37 °C and fixed with cold methanol (pre-chilled at −20 °C).

## Immunofluorescence

For immunofluorescence analysis, the cells were grown on glass coverslips coated with poly-L-lysine. The harvested cells were fixed with pre-chilled methanol at −20 °C for 15 min. Following fixation, the cells were washed with PBS and blocked with PBS-BT (3% BSA, 0.1% Triton X-100, phosphate buffer saline) solution for 30 min. Subsequently, the cells were incubated with primary antibodies diluted in PBS-BT overnight at 4 °C unless specified. The cells were incubated with primary antibodies against ch-TOG (#1, 1:100, Abcam, ab86073), ch-TOG (#2, 1:250, Santa Cruz Biotechnology, sc-374394), CEP128 (1:250, Bethyl, A303-348A), GFP (1:500, Thermo Fisher Scientific, A-6455), ODF2 (1:500, Abcam, ab43840), Ac-α-Tubulin (1:500, SIGMA, T6793), γ-Tubulin (1:500, EXBIO, 11-4645-C100), Ninein (1:500, EMD Millipore Corp, MABT29), PCNT (1:500, Tim Stearns, Stanford University, USA[27]), NEDD1 (1:500; Tim Stearns, Stanford University, USA[27]), α-Tubulin (1:500, SIMGA T6199), α-Tubulin (1:250, Abcam, ab18251). For the cells expressing the GFP-tagged recombinant proteins, anti-GFP antibody immunostaining was performed for strictly 30 min. Following primary antibody incubation, the cells were washed thrice with PBS and incubated with Alexa 488 (1:500), Alexa 568 (1:500), and/or Alexa 647 (1:500) conjugated anti-rabbit or anti-mouse antibodies (Thermo Fisher Scientific) at room temperature (RT) for 1 h. Finally, the cells were washed thrice with PBS and mounted using using ProLong Gold Antifade (Thermo Fisher Scientific) on glass slides.

## Immunoprecipitation and pull-downs

Immunoprecipitation: U2OS cells were lysed in lysis buffer (20 mM Tris-HCl pH 8.0, 100 mM NaCl, 1 mM EGTA, 1 mM MgCl₂, 0.5% NP40, 1 mM PMSF, and protease inhibitor cocktail). Endogenous ch-TOG, GCP3, and NEDD1 were independently immunoprecipitated using antibodies against ch-TOG (Abcam, ab86073), GCP3 (Jens Lüders, IRB Barcelona, Spain)[26], or NEDD1 (Tim Stearns, Stanford University USA)[27], respectively (1 μg of antibody per sample). Unspecific rabbit IgG was used for control immunoprecipitations. The immunoprecipitation was performed for 30 min at 4 °C and later subjected to three washes with lysis buffer. Subsequently the antibody complexes were precipitated using Protein G coupled dynabeads (Thermo Fisher Scientific, #10003D) for a period of 30 min. The collected beads were washed

thrice with lysis buffer, boiled in SDS loading buffer and subjected to analysis by Western blot.

Anti-FLAG and streptavidin pull-downs: HEK293T cells were transfected with plasmid expressing FLAG-tagged BirA alone (Control), GCP2-BirA, or BirA-GCP3 using 8 μg/mL of polyethylenimine (PEI). After 48 h of transfection cells were lysed as above and subjected to pulldown using Anti-FLAG conjugated agarose beads (SIGMA, A2220). For the biotinylation assay, 24 h after transfection cells were incubated with biotin (IBA GmbH, 2-1016-002) at a final concentration of 50 μM for a period of 24 h. Subsequently, the cells were lysed in SDS-lysis buffer (50 mM Tris-HCl pH 8.0, 150 mM NaCl, 0.1% SDS, 1% Triton X-100, 1 mM EDTA, 1 mM EGTA, and protease inhibitor cocktail) for 1 hour and sonicated. Streptavidin conjugated sepharose beads (GE Healthcare, GE17-5113-01) were added to the lysate to bind biotinylated proteins for a period of 3 h at 4 °C. The collected beads were washed thrice using SDS-lysis buffer, boiled in SDS loading buffer and subjected to analysis by Western blot.

## Western blots

The samples were subjected to SDS-PAGE and transferred to polyvinylidene fluoride (PVDF) membrane (Immobilon-P IPVH00010) by tank blotting. The transferred membranes were immunoblotted with antibodies against the following proteins: ch-TOG (#1, 1:1000, Abcam ab86073), ch-TOG (#2, 1:1000, Santa Cruz Biotechnology sc-374394), γ-Tubulin (1:3000, SIGMA T6557), GFP (1:2000, Torrey Pines Biolabs TP401), FLAG (1:10000, SIGMA F1804), GCP3 (1:2000; Jens Lüders, IRB Barcelona, Spain[26]), NEDD1 (1:2000; Tim Stearns, Stanford University, USA[27]), GAPDH (1:10000, Santa Cruz Biotechnology sc-47724). As secondary antibodies HRP-coupled goat anti-rabbit and goat anti-mouse antibodies were used (1:5000; Jackson ImmunoResearch Laboratories, AB_10015289, AB_2313567). The proteins were detected using the SuperSignal West Pico PLUS chemiluminescent substrate kit (Thermo Fisher Scientific).

## Microscopy, image analysis, and intensity quantifications

Three-dimensional structured illumination microscopy (3D-SIM) was performed on a super resolution microscope Elyra PS.1 (Carl Zeiss, Germany). The Z-axis projection slices/images (either 256 × 256 or 512 × 512 frame sizes) were acquired using immersol 518 F oil (Zeiss) on Alpha Plan Apochromat 100x/1.46NA Oil Dic M27 objective lens. The fluorescent dyes Alexa Flour 488, 568, and 647 were excited using 488 nm (20% of 200 mW laser source), 561 nm (10% of 200 mW laser source), and 642 nm (5% of 500 mW laser source) lasers. The emitted light was collected through 495–575 nm (488 nm excitation), 570–650 nm (561 nm excitation), and 655 nm-above (642 excitation) emission filters. The acquired images were subsequently processed using ZEN black software (ZEISS). Further image analysis was performed using Fiji (ImageJ)[83] software. The corresponding Z-axis slices/

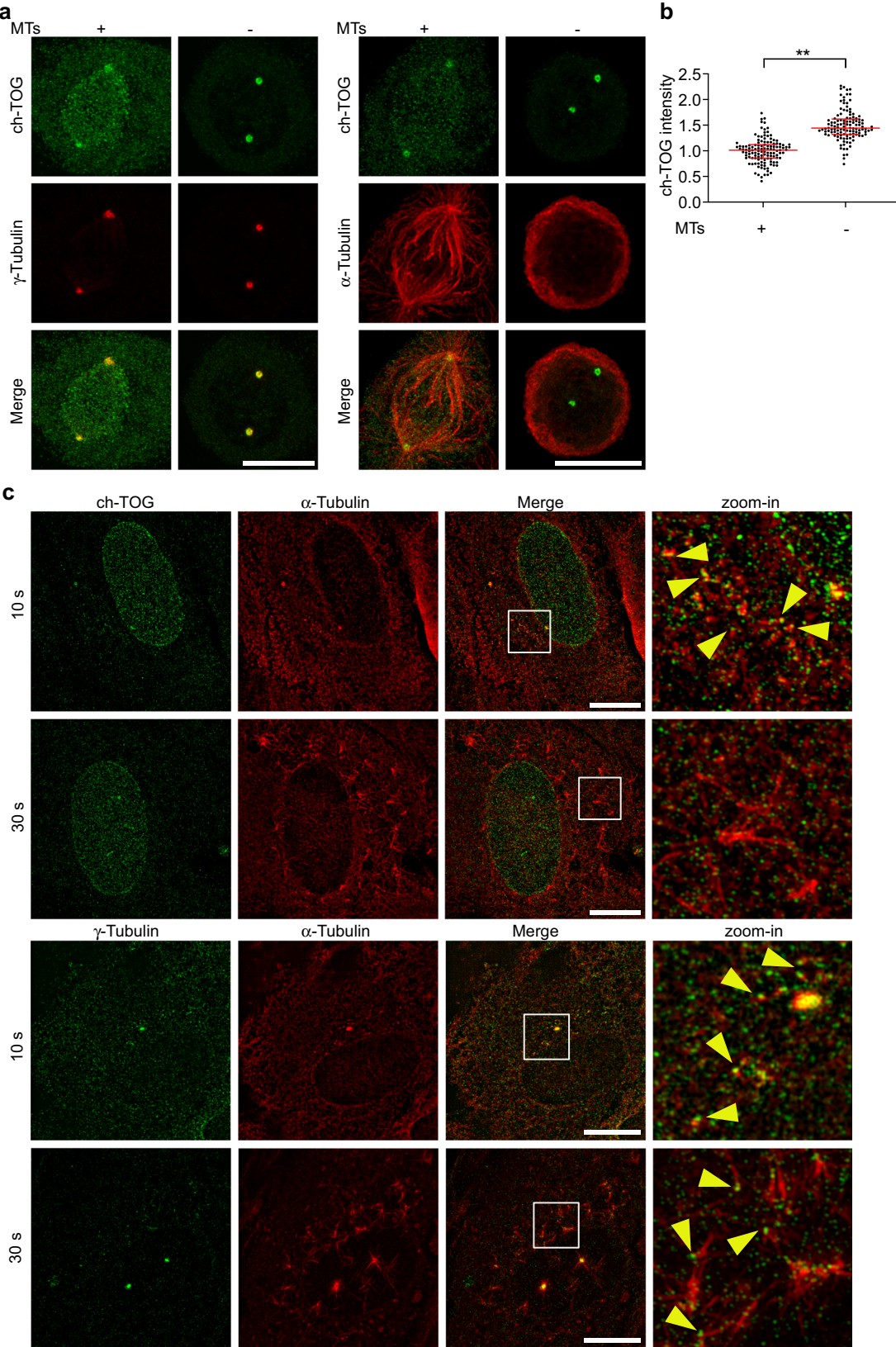

images were subjected to maximum intensity Z-projections for figure panels and sum projections for quantifications using Fiji.

Green and red colors in the immunofluorescence images correspond to Alexa 488 and 568 labels used in the respective channels, white corresponds to Alexa 647. Exceptions: in Supplementary Fig. 2B, chTOG staining (pseudocolored in green) was acquired with Alexa 647;

in Supplementary Fig. 1D, γ-tubulin and pericentrin stainings (pseudocolored in green and red, respectively) were acquired with Alexa 568 and Alexa 488, respectively.

Intensity quantifications of entire centrosomes were made using a 1 μm × 1 μm region of interest (ROI) around the centrioles. Microtubule asters marked by α-tubulin staining were quantified using a

**Fig. 6 | ch-TOG transiently associates with different nucleation sites.**
**a** Maximum intensity projections of 3D-SIM images of ch-TOG localization at mitotic centrosomes in the presence or absence of microtubules. Cells were costained with antibodies against ch-TOG in combination with antibodies against γ-tubulin or α-tubulin. Scale bars, 10 μm. **b** Centrosomal ch-TOG intensities were quantified, normalized to the average of the intensities of the control, and plotted. *N* = 3 experiments, total number of cells analyzed: 63 (MTs+) and 61 (MTs−). \**p* = 0.0129. The horizontal bars and whiskers indicate median and interquartile range, respectively, of the plotted data points. Statistical significance was

determined by unpaired, two-tailed *t* test with Welch's correction. **c** RPE1 cells were subjected to microtubule depolymerization using nocodazole followed by cold treatment. Microtubule regrowth was allowed for 10 or 30 s at 37 °C. Cells were fixed, stained with antibodies against either ch-TOG or γ-tubulin and co-stained with α-tubulin to detect microtubules, and analyzed by 3D-SIM. Yellow arrowheads in zoom-in panels indicate microtubules that have ch-TOG or γ-tubulin signal at one of their ends. Three independent experiments (two stained for ch-TOG/α-tubulin, one stained for γ-tubulin/α-tubulin). Scale bars, 10 μm. Source data are provided as a Source Data file.

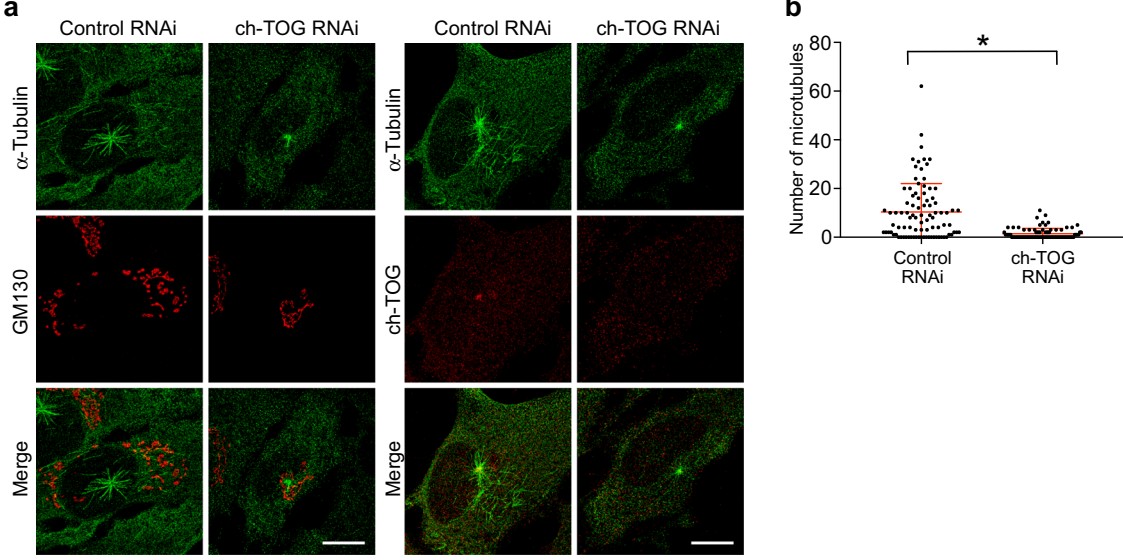

**Fig. 7 | ch-TOG promotes nucleation from the Golgi. a** RPE1 control and ch-TOG RNAi cells were treated with nocodazole to depolymerize microtubules. After washout and incubation in ice, microtubules were allowed to regrow for 10 s. Cells were fixed and stained with antibodies against GM130 (Golgi), α-tubulin (microtubules), and ch-TOG. Scale bar, 10 μm. **b** The number of microtubules growing from the Golgi in control and ch-TOG RNAi cells as in (**a**) were scored and plotted.

*N* = 3 experiments, total number of cells analyzed: 90 in each condition, \**p* = 0.0207. The horizontal bars and whiskers indicate median and interquartile range, respectively, of the plotted data points. Statistical significance was determined by unpaired, two-tailed *t* test with Welch's correction. Source data are provided as a Source Data file.

3.05 μm × 3.05 μm ROI. γ-Tubulin signals at mitotic centrosomes in Supplementary Fig. 5 were quantified in a 2 μm × 2 μm ROI. For PCM and SDA intensity quantifications in Fig. 2c we used rectangular ROIs of 1 μm × 0.69 μm and 1 μm × 0.3 μm, respectively. Using centrioles in side-views, ROIs for PCM and SDA were placed over the proximal and distal centriole ends, respectively, with the ROI rectangles positioned perpendicular to the centriole axis. Mean intensities were measured within ROIs. For quantification of PCM and luminal γ-tubulin intensities in Supplementary Fig. 3, overall centrosomal signal in a 1 μm × 1 μm ROI, and a ROI drawn around the luminal signal were measured as integrated densities. The PCM-specific signal was obtained by subtracting the luminal from the overall signal. All ROI intensities were subjected to background correction from an adjacent area using dimensionally equivalent ROIs. All the absolute intensities were normalized to the average of the control intensities and plotted as dots in a scatterplot using GraphPad Prism software.

To quantify Golgi microtubule regrowth, for each cell the number of microtubules growing from the Golgi at 10 s regrowth was counted in control and ch-TOG depleted cells.

### Quantification and statistical analysis
GraphPad Prism 9.4.0 was used to perform statistical analysis. Statistical significance was determined by unpaired, two-tailed *t* test with Welch's correction. Additional details are found in the figure legends.

### Reporting summary
Further information on research design is available in the Nature Portfolio Reporting Summary linked to this article.

### Data availability
All data generated and analyzed in this study are included in the article and the supplementary information files. Source data are provided with this paper.

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

## Acknowledgements

We are grateful to Anna Akhmanova (Utrecht University, The Netherlands) for providing RPE1 CDK5RAP2 KO cells. We thank Joel Paz for cloning the BirA-GCP3 plasmid. J.L. acknowledges support by grants BFU2015-69275-P (MINECO/FEDER), PGC2018-099562-B-I00 (MICINN), network grants 2017 SGR 1089 (AGAUR) and RED2018-102723-T (MICINN), and by intramural funds of IRB Barcelona, recipient of a Severo Ochoa Centre of Excellence Award from the Spanish Ministry of Science and Innovation and supported by CERCA (Generalitat de Catalunya). A.A. was supported by a Human Frontier Science Program (HFSP) postdoctoral long-term fellowship (Reference No: LT000181/2018-L). C.V. was supported by a doctoral fellowship (PRE2018-083390; MICINN). We thank the IRB Barcelona Advanced Digital Microscopy Facility for excellent support.

## Author contributions

A.A. designed and performed most of the experiments, cloned ch-TOG constructs, analyzed data, performed quantifications, prepared figures, drafted the manuscript, and contributed to editing. C.V. performed clonings, proximity biotinylations, and pull-downs, analyzed centrosomal γ-tubulin recruitment in CDK5RAP2 KO cells and in mitotic cells, measured nucleation from the Golgi, and contributed to figure preparation and manuscript editing. C.L. helped with cloning and initially identified ch-TOG as proximity interactor of GCP2. J.L. acquired funding, supervised the study, proposed experimental strategies, and contributed to manuscript writing and figure editing.

## Competing interests

The authors declare no competing interests.
