## [Peer Review File · Nature Communications]

Microtubule nucleation and γ TuRC centrosome localization in interphase cells require ch-TOGREVIEWER COMMENTS

Reviewer #1 (Remarks to the Author):

In CKAP5-dependent localization and activity of the microtubule nucleator gTuRC Ali et al. present extensive imaging and molecular evidence about the role of CKAP5 in microtubule formation, particularly around the centriole. Overall this is a good paper which presents a clear story with extensive evidence. The authors argue their interpretation well and to come to solid conclusions about the biological pathway in question.

I was particularly impressed with the clearly stated statistics. Every comparison has clear experimental repeats, N numbers and significance with a stated statistical test. All box and whisker graphs had the same format which was again stated in every legend.

Major issues:

I have one major reservation about this paper. The paper relies almost exclusively on image data and yet there is NO information about the microscopy performed except that it is "3DSIM". The methods sections clearly states what antibodies used, even which dyes were used, yet they never state what microscope, what objective, what optical xyz resolution, or pixel resolution, what excitation wavelengths, emission wavelengths, or what protein was labeled with which dye in any given image. This is a criminal lack of methods information, particularly with the recent focus on improving microscopy meta data and reproducibility (see virtually the whole of Nature Methods Dec 2021).

Minor issues:

Results section line 127: The authors claim that the CKAP5 only interacts with one of the two centrioles and yet in the panel labeled Antibody 2 in fig 1A it is clearly interacting in a similar manner with both centrioles.

In the next sentence the authors claim that the CKAP5 "colocalized" with the NIN marker. I have 2 issues with this, 1) at super resolution the term co-localized becomes increasingly un-informative and 2) in the panel in question of fig 1A there is maybe 10% overlap between the signals in the subdistal region they are referring to. This is not colocalized.

Line 151: "Careful analysis revealed that a small amount of g-tubulin was also detectable at subdistal 153 appendages, indicating that colocalization of CKAP5 and gTuRC occurred at both sites 154 (Fig. 2A, white arrow heads; Supp. Fig. 1D)." There is definitely signal here but I am not convinced it is not bleed through. It is in the very brightest region of the signal colour green in the figure and with no information about dyes, illumination or filters it is impossible to

tell if this might not be a few % of bleed through signal.

Paragraph at 163: The repolymerization assay is a nice experiment but I am totally unconvinced by their time course. How much happens while they are trying to fix their sample? The authors either need to do a time lapse image series, which would be possible in widefield as the SIM resolution is not required, but would require using different tagging strategy rather than their anti-body labeling. An alternative minimum would be to have a +0s time point where the regrowth is triggered the cells immediately fixed.

Text around 222 and fig 4, I am not at all convinced that the microtubules grow from the subdistal appendages. The images are not of sufficiently high resolution and it is nowhere near possible to follow individual microtubules. I think this is an interesting idea but the current data do not support (or refute) it.

In figure 5C the green coloured panels with CKAP5 RNAi seem to universally have significantly more background than those with the control RNAi, this makes some of the claims in lines 250-260 hard to justify. Is there really a difference between the recruitment with CKAP5-345C, or are we just confused with the much higher background?

line 353: "...CEP192 and NEDD1 were found to be in close proximity at centrosomes by biotin proximity labeling." I think this is a bit of a stretch, the biotin experiments show that the proteins were in close proximity but not necessarily at the centrosome. It is likely that it is occurring at the centrosome but I don't think the experiments show this.

Line 475: no information about Alex647 antibody dilution are given.

line 483: no information is given about how different regions were segmented to make these measurements.

line 486: How was background correction performed, this is more important than the software used.

Reviewer #2 (Remarks to the Author):

In their manuscript, the authors Ali et al. utilize a high-resolution imaging technique (3D-SIM) to investigate the role of the MT polymerase, CKAP5/ch-TOG, in the recruitment of the gamma-tubulin ring complex (γ TuRC) to nucleation sites in human cells. Notably, the authors find that RNAi depletion of CKAP5/ch-TOG leads to a roughly 50% decrease in gamma-tubulin recruitment to centrosomes in human cells (Fig. 3B). Furthermore, the authors use biotin-proximity labeling and detect a transient interaction between human γ TuRC and human CKAP5/ch-TOG. Furthermore, the authors find that depletion of CKAP5/ch-TOG results in loss of MT nucleation from the PCM. Lastly, exogenous expression of CKAP5/ch-TOG truncation constructs demonstrate that the full-length protein is required for both MT nucleation and full recruitment of γ TuRC to MTOCs.

The main technique used in this work, 3D-SIM, is well-established and has been used before to look at the distribution of centrosomal components (Lawo et al., Nat Cell Biol, 2012; Mennella et al., Nat Cell

Biol, 2012). Its use here is appropriate for the question being investigated. However, it lacks the fast time-scale of established assays at the single molecule level for analogous studies of microtubule nucleation (Roostalu et al., *Elife* 2015; Thawani et al., *Elife* 2020; Consolati et al, *Dev Cell* 2020), in which transient interaction of ch-TOC/XMAP215 with γ TuRC had already been reported.

While classically known only as MT polymerases, CKAP5/ch-TOG's homologs, XMAP215 and Alp14, have previously been reported as essential components for MT nucleation with γ TuRC in fission yeast and frogs (Gunzelmann et al., *Elife* 2018; Flor-Parra et al. *Curr Biol.*, 2018; Thawani et al. *Nat Cell Biol.*, 2018). As such, the authors' findings validate these prior works and extend them to multiple human cell lines. Similarly, the truncation constructs tested in this work validate similar work by Thawani et al, *Nat Cell Biol.*, 2018, demonstrating two distinct regions within XMAP215/ch-TOG: an N-terminal tubulin binding region necessary for polymerization and a C-terminal gamma-tubulin binding region necessary for γ TuRC interaction.

The main novelty is that CKAP5/ch-TOG has a function in recruiting γ TuRC, and not only synergistically nucleating microtubules with it. However, the major points below need to be addressed in order to support this claim.

Major points

1) The authors should use the better-known and widely-used designation of ch-TOG, instead of CKAP5, to maintain continuity with the prior literature. Switching to the name CKAP5 makes it seem like a new protein/function discovery, which is not the case.

2) While the data broadly support an essential role for CKAP5/ch-TOG in MT nucleation, the authors' assertion that "at centrosomes...CKAP5 ultimately controls γ TuRC recruitment and activation" (lines 17-18) is less convincing. The authors state that "in the absence of CKAP5, γ TuRC fails to localize to centrosomal nucleation sites," (lines 20-21) but the 3D-SIM data shows that complete CKAP5 depletion only results in roughly 50% loss of gamma-tubulin signal from the centrosome (Fig. 3A,B). This suggests that additional mechanisms, like the known γ TuRC binding partners (CDK5RAP2, myomegalin, CEP192) play a redundant role. The text should be re-written to acknowledge that even complete loss of ch-TOG does not result in the complete failure of centrosomal γ TuRC recruitment. This in itself leads to the key "chicken or egg" question: If ch-TOG is 50% depleted, one would expect less microtubule nucleation and therefore less γ TuRC. Can the authors clearly differentiate between this possibility and the conclusion they draw, i.e. ch-TOG recruits γ TuRC?

3) Prior work by the Qi lab has shown that overexpression of the CM1 domain from CDK5RAP2 can trigger redistribution of 90% of γ TuRCs from the centrosome to ectopic sites (Fong, Ka-Wing et al., *Mol Biol Cell*, 2008). In that same study, Fong et al also showed that "severe knockdown of CDK5RAP2 yielded almost complete loss of gamma-tubulin at the centrosomes," while complete loss of CKAP5/ch-TOG in this work only results in 50% loss of gamma-tubulin signal. With this in mind, why do the authors conclude that CKAP5/ch-TOG is the dominant recruitment factor for γ TuRC to the centrosome? ch-TOG depletion should be compared to CDK5RAP2 knock-down to assess ch-TOG recruitment ability (related to comment 2).

4) Why do the authors conclude that "stimulation of microtubule nucleation" is "crucial for stable MTOC incorporation of γ TuRC" (lines 326-327) when other studies have found that centrosomal adapters can independently bind and anchor γ TuRC? For example, the CM1 motif binds γ TuRC strongly enough to allow purification and subsequent structure determination (Wieczorek et al., *Cell*, 2020). Such a stable interaction does not appear to require additional factors and suggests CKAP5/ch-TOG is not required. Furthermore, the authors suggest that "nucleation of a microtubule, stimulated by CKAP5 [ch-TOG], may induce a conformational change in γ TuRC that promotes its binding to MTOC-bound adapters (lines 327-329)," but offer no evidence for this suggestion. The existing literature suggests that adapters like CDK5RAP2 do not require ch-TOG to bind γ TuRC. This could easily be tested via the experiment suggested in comment 3.

5) Biotin proximity labeling to detect transient γ TuRC interactions has previously been used by the Lüders group (Schweizer et al, *Nat Comm*, 2021) and is used here to assay for ch-TOG (Fig. 3F,G). While their most recent use of this assay benefitted from an unbiased mass-spectrometry analysis of co-precipitants, this work directly probes for ch-TOG, without including controls to assess whether this

interaction is above the background level of biotinylation. The authors should probe for proteins known not to bind γ TuRC for comparison against the ch-TOG band.

a. Furthermore, there is a discrepancy in the gamma-tubulin signal in the FLAG-GCP3-BirA pulldowns in Fig. 3G and the FLAG-GCP2-BirA pulldowns in Suppl. Fig. 3D. The gamma-tubulin signal in the GCP2 pulldowns is faint, while the band is stronger in the GCP3 pulldowns. How do the authors explain this difference?

Minor points

6) Why was the role of the known CKAP5/ch-TOG partner, TACC3 not investigated? TACC3 is known to interact with the C-terminus of CKAP5/ch-TOG and is known to be important for its different functions at the centrosomes and in the mitotic spindle (Thakur et al., J Biol Chem, 2014). TACC3 could be an illuminating target for further study.

7) Line 179: "strong overall decrease"; Figure 3B indicates this is a 50% decrease. Please use more specific language (e.g. 50% decrease) instead of "strong overall decrease."

8) Line 188: "strongly reduced"; same issue as point 7.

9) Lines 221-222: "Some microtubules..."; same issue as point 7.

10) Lines 230-231: "...some regrown microtubules coincided with NIN..."; same issue as point 7.

Reviewer #3 (Remarks to the Author):

In this paper, the authors examine the localisation of the human TOG domain protein, CKAP5, and the effect of its depletion on γ -TuRC localisation and microtubule nucleation at interphase centrosomes in cultured cells. They find that CKAP5 localises predominantly to sub-distal appendages, but also localises transiently to the interphase PCM – localisation to the interphase PCM is observed only after microtubule depolymerisation and depends upon the c-terminal (non-TOG domain) region – consistent with this, they identify a transient interaction between CKAP5 and γ -TuRC via a bio-ID assay, but do not observe a stable interaction via an IP assay. They also show that depletion of CKAP5 leads to a reduction in γ -TuRC localisation to the interphase PCM and to reduced nucleation. Most studies of TOG domain proteins have either been performed in vitro or on CKAP5 homologues and so it is refreshing to see a study on human CKAP5 (also known as ch-TOG) within cells. Nevertheless, much of what has been observed in this study is already known for TOG proteins in general i.e. they can interact with γ -TuRC and promote microtubule nucleation. New insight, however, comes from the observation that CKAP5 is required for γ -TuRC localisation to the interphase PCM, which is an unexpected result. It remains unclear why this is the case and whether it is specific to the interphase PCM. Indeed, the authors mention that a previous paper showed there was no effect on γ -tubulin levels at mitotic centrosomes after CKAP5 depletion. It would have been nice if the authors had confirmed this here or shown that there is in fact a quantifiable effect at mitotic centrosomes – it seems like a very doable experiment. Overall, the results are solid and presented clearly but we do not learn a great deal about how CKAP5 functions. It should be an editorial decision as to whether they feel the paper is suitable for publication in Nature Communications. I have a few comments listed below that the authors should address.

1) I think it is important that the text reflects the fact that the authors have studied CKAP5 only at interphase centrosomes. There are examples of differences between interphase and mitotic centrosomes in the literature, and the centrosome is only one of many different MTOCs. The current title is "CKAP5-dependent localization and activity of the microtubule nucleator γ TuRC". I don't think the authors provide any data regarding the activity of γ -TuRCs, but rather show that CKAP5 is required for localisation of γ -TuRC to interphase centrosomes. A more apt title would be "CKAP5-dependent localization of the microtubule nucleator γ TuRC at interphase centrosomes". In the abstract the authors say: "Here we show that at centrosomes, rather than adapters, the microtubule polymerase CKAP5 ultimately controls γ TuRC recruitment and activation". They should say "Here we show that at interphase centrosomes, rather than adapters, the microtubule polymerase CKAP5

ultimately controls gTuRC recruitment and activation". Also "In the absence of CKAP5, gTuRC fails to localize to centrosomal nucleation sites" should read "In the absence of CKAP5, gTuRC fails to localize to centrosomal nucleation sites during interphase". And "Together our data show that centrosome attachment of gTuRC and activation of nucleation are mechanistically coupled through transient recruitment of CKAP5" should be "Together our data show that attachment of gTuRC at interphase centrosomes are mechanistically coupled through transient recruitment of CKAP5". In Discussion the authors say: "Recruitment and activation of the microtubule nucleator gTuRC are fundamental to the function of MTOCs. In this study we show that at centrosomes in human cells the XMAP215 family member CKAP5 is crucial for both of these functions", when this should be "In this study we show that at interphase centrosomes in human cells the XMAP215 family member CKAP5 is crucial for γ -TuRC recruitment". Also in the Discussion the authors say: "In contrast, our results indicate that in human cells CKAP5 is crucial for stable MTOC incorporation of gTuRC through a mechanism that involves stimulation of nucleation". The authors cannot generalise to all MTOCs. Then they say "This is further supported by our finding that CKAP5 centrosome localization, in turn, also depends on gTuRC" – needs to be ...centrosome localization to interphase centrosomes.

2) Supp Fig 1A – the authors say that "Only in late G2 phase CKAP5 was also detected at more proximal centriole regions, in addition to its localization to subdistal appendages (Supp. Fig. 1A)" but I find it difficult to see a difference in the localisation pattern through the cell cycle from the images shown. Perhaps the authors can point these differences out more clearly.

3) It would be useful for the authors to quantify CKAP5 levels at centrosomes after CKAP5 RNAi.

4) How were intensity measurements made exactly? Are they mean intensity measurements or sum intensity measurements?

5) The authors say that CKAP5 and γ -tubulin colocalised: "CKAP5 accumulation occurred mostly in the proximal centriole region, where it colocalized with γ -tubulin suggesting that it was present in the PCM (Fig. 2A, white arrows; Supp. Fig. 2B)", but it seems that the signals do not perfectly overlap. This could probably best be seen with line scans. I think the authors should check this carefully and re-phase the sentence if necessary.

6) The experiment in Fig 6C is not cited in the main text. Moreover, it is unclear where the non-centrosomal microtubules are originating from – are these from the Golgi, as the authors mentioned that these cells nucleate some microtubules from the Golgi? Or are they forming randomly in the cytoplasm? If the authors can determine this, they can then expand their analysis to non-centrosomal sites e.g. they could assess microtubule nucleation from these sites after CKAP5 knockdown. This would expand the breadth of the paper outside of just interphase centrosomes.

Response to reviewers' comments

Reviewer #1:

In CKAP5-dependent localization and activity of the microtubule nucleator gTuRC Ali et al. present extensive imaging and molecular evidence about the role of CKAP5 in microtubule formation, particularly around the centriole. Overall this is a good paper which presents a clear story with extensive evidence. The authors argue their interpretation well and to come to solid conclusions about the biological pathway in question. I was particularly impressed with the clearly stated statistics. Every comparison has clear experimental repeats, N numbers and significance with a stated statistical test. All box and whisker graphs had the same format which was again stated in every legend. We thank the reviewer for the overall positive evaluation and useful feedback. Our detailed responses to specific points are below.

Major issues:

I have one major reservation about this paper. The paper relies almost exclusively on image data and yet there is NO information about the microscopy performed except that it is "3DSIM". The methods sections clearly states what antibodies used, even which dyes were used, yet they never state what microscope, what objective, what optical xyz resolution, or pixel resolution, what excitation wavelengths, emission wavelengths, or what protein was labeled with which dye in any given image. This is a criminal lack of methods information, particularly with the recent focus on improving microscopy meta data and reproducibility (see virtually the whole of Nature Methods Dec 2021).

We apologize for this omission. We have added this information to the method section.

Minor issues:

Results section line 127: The authors claim that the CKAP5 only interacts with one of the two centrioles and yet in the panel labeled Antibody 2 in fig 1A it is clearly interacting in a similar manner with both centrioles.

The reviewer is right. We noticed differences in the staining with two CKAP5 antibodies and the text did not accurately describe this. The difference concerns a signal that is only seen with antibody #2. This signal is observed at the distal end of both centrioles. In the case of the mother, when viewed from the side, the signal is positioned between the subdistal appendage labeling. At the daughter it is observed without subdistal appendage signal, since these are not present. We know that it is the distal end through co-staining of centrioles with acetylated tubulin, which is always slightly weaker at the distal end. The CKAP5 distal end signal is not seen with antibody #1, which we have used throughout the paper, but only with antibody #2. Importantly, these differences do not affect the data presented in this manuscript: both antibodies label CKAP5 at the subdistal appendages and show CKAP5 accumulation around more proximal centriole regions upon microtubule

depolymerization. Future work needs to determine the significance of the CKAP5 distal end localization. We have changed the text and added labels to Fig. 1A, to describe the localization more accurately, and have added additional stainings with antibody #2, in the presence and absence of microtubules (Supp. Fig. 1A)

In the next sentence the authors claim that the CKAP5 "colocalized" with the NIN marker. I have 2 issues with this, 1) at super resolution the term co-localized becomes increasingly un-informative and 2) in the panel in question of fig 1A there is maybe 10% overlap between the signals in the subdistal region they are referring to. This is not colocalized. We agree that colocalization may only be partial. Also, as the reviewer suggests, at super resolution proteins within the same structure (e.g. sub-distal appendages) may be visibly present in different sub-compartments. We have rephrased the statement and now say: "...where its distribution resembled subdistal appendage localization. Indeed, CKAP5 partially colocalized with the subdistal appendage marker NIN (ninein)."

Line 151: "Careful analysis revealed that a small amount of γ -tubulin was also detectable at subdistal 153 appendages, indicating that colocalization of CKAP5 and γ TuRC occurred at both sites 154 (Fig. 2A, white arrow heads; Supp. Fig. 1D)." There is definitely signal here but I am not convinced it is not bleed through. It is in the very brightest region of the signal colour green in the figure and with no information about dyes, illumination or filters it is impossible to tell if this might not be a few % of bleed through signal.

As indicated above, we have included information about the hardware setup in the method section. From our experience with this setup we can exclude channel bleed-through as source of the signal. This is very clear in the stainings of Supp. Fig. 1D, which we already included and referred to in the original manuscript. Here we have co-stained γ -tubulin with pericentrin, which does not produce any signal at subdistal appendages. Still, the signal of γ -tubulin in the subdistal appendage region is clearly observed.

Paragraph at 163: The repolymerization assay is a nice experiment but I am totally unconvinced by their time course. How much happens while they are trying to fix their sample? The authors either need to do a time lapse image series, which would be possible in widefield as the SIM resolution is not required, but would require using different tagging strategy rather than their anti-body labeling. An alternative minimum would be to have a +0s time point where the regrowth is triggered the cells immediately fixed. We note the "0 s" time point was already included in the original figure. To make this clearer, we have improved the labeling of the figure panel. Our shortest regrowth time point is 5 s, which is just enough time for a few short microtubules to form.

To further address the reviewer's concern regarding the precision of the time course, we have added a more detailed description including a protocol reference (Ezquerro et al., 2020) for the regrowth assay in the methods. Briefly, in this assay we position culture dishes with cells on coverslips in an ice-water bath right next to pre-warmed medium held in a 37°C water bath and a dish with ice-cold methanol. We grab coverslips with forceps from the dish with ice-cold medium and immerse these directly in methanol ("0 s"). Contrary perhaps to fixation in formaldehyde buffer, fixation in methanol is immediate. Once immersed in ice-cold methanol, there is no microtubule growth anymore. This is also illustrated by the absence of microtubules in this condition (Supp.

Fig 1E). For regrowth, we grab coverslips from the ice-cold medium and immerse them immediately in pre-warmed medium, hand-holding them there for the indicated time, before immediately transferring them to ice-cold methanol for fixation. The time between immersions in the different conditions is minimal (~1 s). We hope that this convinces the reviewer that our methodology is precise and reliable.

Finally, we would like to emphasize that the most important conclusion from the time course experiment is not the precise timing of events, but rather the fact that the CKAP5 signal is clustered around centrioles at the “0 s” time point and becomes dispersed in association with microtubule plus ends as soon as microtubules start growing from centrioles.

Text around 222 and fig 4, I am not at all convinced that the microtubules grow from the subdistal appendages. The images are not of sufficiently high resolution and it is nowhere near possible to follow individual microtubules. I think this is an interesting idea but the current data do not support (or refute) it.

We agree that we work at the resolution limit here and that individual microtubules are difficult to resolve in dense asters. Still, it can be seen that microtubule regrowth occurs not only from the PCM region, but also from more distal sites. In particular in RNAi conditions with strongly reduced nucleation (Fig. 4A), some of the few microtubules that do form, clearly originate from the distal appendage region. To illustrate this better, we have added additional arrowheads and magnifications also to Fig. 4A. Considering the remaining limitations of the available tools and assays, we have also toned down statements that refer to subdistal appendages as nucleation sites in results and discussion.

In figure 5C the green coloured panels with CKAP5 RNAi seem to universally have significantly more background than those with the control RNAi, this makes some of the claims in lines 250-260 hard to justify. Is there really a difference between the recruitment with CKAP5-345C, or are we just confused with the much higher background?

Indeed, recombinant CKAP5 protein expression levels were slightly increased when endogenous CKAP5 was depleted, as can also be seen in the western blots. We carefully re-examined multiple examples of CKAP5-345C expressing cells and confirmed that the results are as presented in the manuscript: for CKAP5-345C localization to the centrosome was robust in controls, but strongly reduced in CKAP5 RNAi cells, independent of background fluorescence in individual images. We have added an additional example panel of CKAP5-345C cells depleted of endogenous CKAP5 in Fig. 5C to further support this conclusion. In contrast, the CKAP5-C construct, which also showed increased expression and background in cells after CKAP5 RNAi, displayed robust centrosome localization.

line 353: "...CEP192 and NEDD1 were found to be in close proximity at centrosomes by biotin proximity labeling." I think this is a bit of a stretch, the biotin experiments show that the proteins were in close proximity but not necessarily at the centrosome. It is likely that it is occurring at the centrosome but I don't think the experiments show this.

We would like to mention that in contrast to other studies, in the cited paper the authors performed BioID on centrosomal fractions rather than using whole cell extract. However,

since one cannot exclude that proximity labeling may have occurred outside of centrosomes and that the labeled proteins may have been subsequently incorporated at centrosomes, we have removed “at centrosomes” from our statement.

Line 475: no information about Alex647 antibody dilution are given.

The dilution was 1:500. This information was added to the manuscript.

line 483: no information is given about how different regions were segmented to make these measurements.

We have added this information to the method section.

line 486: How was background correction performed, this is more important than the software used.

We have added a description of how background correction was performed to this section.

Reviewer #2:

In their manuscript, the authors Ali et al. utilize a high-resolution imaging technique (3D-SIM) to investigate the role of the MT polymerase, CKAP5/ch-TOG, in the recruitment of the gamma-tubulin ring complex (γ TuRC) to nucleation sites in human cells. Notably, the authors find that RNAi depletion of CKAP5/ch-TOG leads to a roughly 50% decrease in gamma-tubulin recruitment to centrosomes in human cells (Fig. 3B). Furthermore, the authors use biotin-proximity labeling and detect a transient interaction between human γ TuRC and human CKAP5/ch-TOG. Furthermore, the authors find that depletion of CKAP5/ch-TOG results in loss of MT nucleation from the PCM. Lastly, exogenous expression of CKAP5/ch-TOG truncation constructs demonstrate that the full-length protein is required for both MT nucleation and full recruitment of γ TuRC to MTOCs.

The main technique used in this work, 3D-SIM, is well-established and has been used before to look at the distribution of centrosomal components (Lawo et al., Nat Cell Biol, 2012; Mennella et al., Nat Cell Biol, 2012). Its use here is appropriate for the question being investigated. However, it lacks the fast time-scale of established essays at the single molecule level for analogous studies of microtubule nucleation (Roostalu et al., Elife 2015; Thawani et al., Elife 2020; Consolati et al, Dev Cell 2020), in which transient interaction of ch-TOC/XMAP215 with γ TuRC had already been reported.

While classically known only as MT polymerases, CKAP5/ch-TOG's homologs, XMAP215 and Alp14, have previously been reported as essential components for MT nucleation with γ TuRC in fission yeast and frogs (Gunzelmann et al., Elife 2018; Flor-Parra et al. Curr Biol., 2018; Thawani et al. Nat Cell Biol., 2018). As such, the authors' findings validate these prior works and extend them to multiple human cell lines. Similarly, the truncation constructs tested in this work validate similar work by Thawani et al, Nat Cell Biol., 2018, demonstrating two distinct regions within XMAP215/ch-TOG: an N-terminal tubulin binding region necessary for polymerization and a C-terminal gamma-

tubulin binding region necessary for γ TuRC interaction. The main novelty is that CKAP5/ch-TOG has a function in recruiting γ TuRC, and not only synergistically nucleating microtubules with it. However, the major points below need to be addressed in order to support this claim.

We appreciate the reviewer's rigorous assessment. Regarding previous work that implicated XMAP215 family members in nucleation, we would like to add that only the work in yeast was performed in cells, the work on frog XMAP215 was done exclusively in vitro, in the absence of any MTOC. None of the previous studies in mammalian cells have investigated cooperation of XMAP215/ch-TOG/CKAP5 with γ TuRC in nucleation. We believe that this is a crucial knowledge gap, in particular since the mechanism by which γ TuRC becomes an active nucleator in vertebrate cells remains unknown. Moreover, as correctly stated by the reviewer, the finding that γ TuRC centrosome recruitment depends on CKAP5-dependent activation of nucleation was completely unexpected. The result is very important, since it challenges current models for γ TuRC recruitment to centrosomes (and potentially other MTOCs).

Another important aspect is the question whether CKAP5 may cooperate with γ TuRC in nucleation more generally, or only at centrosomes. Our study demonstrates a role for CKAP5 in both centrosomal and non-centrosomal nucleation. In the revised manuscript we now establish that CKAP5 also promotes nucleation at the Golgi (new Fig. 7).

Major points:

1) The authors should use the better-known and widely-used designation of ch-TOG, instead of CKAP5, to maintain continuity with the prior literature. Switching to the name CKAP5 makes it seem like a new protein/function discovery, which is not the case.

We have used the official gene/protein name, as suggested by the Nature Communications author instructions. That being said, we don't mind changing the name. Before doing so, we will consult with the editor, to also comply with journal formatting requirements.

2) While the data broadly support an essential role for CKAP5/ch-TOG in MT nucleation, the authors' assertion that "at centrosomes...CKAP5 ultimately controls γ TuRC recruitment and activation" (lines 17-18) is less convincing. The authors state that "in the absence of CKAP5, γ TuRC fails to localize to centrosomal nucleation sites," (lines 20-21) but the 3D-SIM data shows that complete CKAP5 depletion only results in roughly 50% loss of gamma-tubulin signal from the centrosome (Fig. 3A,B).

In contrast to most previous studies, we distinguish which sub-centrosomal fractions of γ -tubulin are affected in our depletion experiments. As explained in the results, while overall reduction of the CKAP5 signal at centrosomes is about 50%, in Fig. 3A we show that γ -tubulin is almost completely removed from the outside of centrioles (where nucleation sites are located). The remaining signal is mostly from γ -tubulin in the centriole lumen (where it does not nucleate, but promotes centriole stability, as we recently showed) (Schweizer et al., 2021). Thus, we believe that our statement in the abstract, which refers

to nucleation sites on the outside of centrioles, is correct. We now mention in the abstract that luminal γ TuRC is unaffected, to avoid confusion.

This suggests that additional mechanisms, like the known γ TuRC binding partners (CDK5RAP2, myomegalin, CEP192) play a redundant role. The text should be re-written to acknowledge that even complete loss of ch-TOG does not result in the complete failure of centrosomal γ TuRC recruitment.

As explained in the previous point, in CKAP5-depleted cells the remaining centrosome signal of γ -tubulin is mostly from the centriole lumen, where γ TuRC is recruited by POC5 and augmin to promote centriole integrity, not nucleation (Schweizer et al., 2021). The loss of γ -tubulin from the outside of centrioles suggests that in the absence of CKAP5, the known γ TuRC attachment factors such as CDK5RAP2 or CEP192, cannot recruit significant amounts of γ TuRC. This is one of the important findings of our study that challenge our current understanding of how γ TuRC attachment factors work. Regarding the specific role of CDK5RAP2, see also our responses below.

This in itself leads to the key “chicken or egg” question: If ch-TOG is 50% depleted, one would expect less microtubule nucleation and therefore less γ TuRC. Can the authors clearly differentiate between this possibility and the conclusion they draw, i.e. ch-TOG recruits γ TuRC?

We do not claim that CKAP5 attaches γ TuRC at centrioles like other attachment factors do. It cannot function in this way, since it is not permanently associated with the outside of centrioles. As we state in the manuscript, transient CKAP5 recruitment and activation of nucleation are the requirements for γ TuRC binding to the outside of centrioles, where it then may get more stably attached via known adapter proteins. Indeed, we showed previously that CEP192 depletion, similar to CKAP5 depletion in the current study, efficiently removes γ TuRC from the outside of interphase centrioles, without affecting the luminal pool (Schweizer et al., 2021).

3) Prior work by the Qi lab has shown that overexpression of the CM1 domain from CDK5RAP2 can trigger redistribution of 90% of γ TuRCs from the centrosome to ectopic sites (Fong, Ka-Wing et al., Mol Biol Cell, 2008). In that same study, Fong et al also showed that “severe knockdown of CDK5RAP2 yielded almost complete loss of gamma-tubulin at the centrosomes,” while complete loss of CKAP5/ch-TOG in this work only results in 50% loss of gamma-tubulin signal. With this in mind, why do the authors conclude that CKAP5/ch-TOG is the dominant recruitment factor for γ TuRC to the centrosome? ch-TOG depletion should be compared to CDK5RAP2 knock-down to assess ch-TOG recruitment ability (related to comment 2).

In our view the importance of CDK5RAP2 for recruiting γ TuRC to interphase centrosomes is controversial. Fong et al. (Fong et al., 2008), despite stating “almost complete loss of gamma-tubulin” as cited by the reviewer, did not provide quantification of centrosomal γ -tubulin after CDK5RAP2 RNAi. Importantly, two more recent studies have independently generated CDK5RAP2 KO cell lines and did not observe strong effects on γ -tubulin recruitment and nucleation activity at interphase centrosomes (Gavilan et al., 2018; Wu et al., 2016). To further address this, we have followed the reviewer’s suggestion and tested γ TuRC centrosome recruitment in CDK5RAP2 KO cells. We found that lack of

CDK5RAP2 results in a minor reduction of γ -tubulin signal at interphase centrosomes compared to wildtype cells. Importantly, in CDK5RAP2 KO cells centrosomal γ -tubulin is further reduced upon CKAP5 RNAi without affecting luminal γ -tubulin, similar to the results in wild type cells. Together the new data presented in Supp. Fig. 3 confirm that only a minor fraction of γ TuRC at interphase centrosomes is recruited by CDK5RAP2 and suggest that the majority of γ TuRCs depend on CKAP5.

4) Why do the authors conclude that “stimulation of microtubule nucleation” is “crucial for stable MTOC incorporation of γ TuRC” (lines 326-327) when other studies have found that centrosomal adapters can independently bind and anchor γ TuRC? For example, the CM1 motif binds γ TuRC strongly enough to allow purification and subsequent structure determination (Wieczorek et al., Cell, 2020). Such a stable interaction does not appear to require additional factors and suggests CKAP5/ch-TOG is not required.

As explained above, at interphase centrosomes CDK5RAP2 seems to be less important for γ TuRC recruitment and activation than suggested by initial work. Regarding the ability of adapters to “independently bind and anchor γ TuRC”, to our knowledge this has not yet been demonstrated for adapters in human cells. Perhaps the best studied interaction occurs via the CM1 motif, which is also present in CDK5RAP2. However, whereas the isolated CM1 motif readily binds γ TuRC, the situation is different in the context of full-length adapters. For example, recent work by the Conduit lab (Tovey et al., 2021) has shown that in *Drosophila* Cnn and human CDK5RAP2 the CM1 motif is flanked by phosphorylation-regulated, “auto-inhibitory” sequences that mask and prevent binding of CM1 to γ TuRC.

Furthermore, the authors suggest that “nucleation of a microtubule, stimulated by CKAP5 [ch-TOG], may induce a conformational change in γ TuRC that promotes its binding to MTOC-bound adapters (lines 327-329),” but offer no evidence for this suggestion. The existing literature suggests that adapters like CDK5RAP2 do not require ch-TOG to bind γ TuRC. This could easily be tested via the experiment suggested in comment 3.

As requested by the reviewer and outlined in more detail in our response to point 3 above, we have addressed this by experiments in CDK5RAP2 KO cells. This showed that CDK5RAP2 had only a minor contribution to centrosomal γ TuRC recruitment and that the CKAP5-dependent mechanism also occurred in the complete absence of CDK5RAP2. Thus, other adapters such as CEP192 are likely involved.

The speculation about a conformational change in γ TuRC that may be triggered by nucleation and that may allow it to bind to adapters is part of our discussion, and compatible with the available data. A conformational change in γ TuRC upon nucleation activation is a reasonable assumption and has been suggested previously (Kollman et al., 2015). Consistently, multiple recent studies revealed that the structure of γ TuRC is intrinsically asymmetric and would require a conformational change to become a good template for nucleation (Consolati et al., 2020; Liu et al., 2020; Wieczorek et al., 2020; Zimmermann et al., 2020). However, observing this conformational change experimentally is a major challenge and is currently pursued by various labs in the field. We hypothesize that CKAP5 could play a role in this mechanism through its nucleation activating function, and this may render γ TuRC competent to bind to centrosomal adapters.

5) Biotin proximity labeling to detect transient γ TuRC interactions has previously been used by the Lüders group (Schweizer et al, Nat Comm, 2021) and is used here to assay for ch-TOG (Fig. 3F,G). While their most recent use of this assay benefitted from an unbiased mass-spectrometry analysis of co-precipitants, this work directly probes for ch-TOG, without including controls to assess whether this interaction is above the background level of biotinylation. The authors should probe for proteins known not to bind γ TuRC for comparison against the ch-TOG band. a. Furthermore, there is a discrepancy in the gamma-tubulin signal in the FLAG-GCP3-BirA pulldowns in Fig. 3G and the FLAG-GCP2-BirA pulldowns in Suppl. Fig. 3D. The gamma-tubulin signal in the GCP2 pulldowns is faint, while the band is stronger in the GCP3 pulldowns. How do the authors explain this difference?

We have repeated the proximity biotinylation experiments and probed for GAPDH as a control protein that is detected in the input, but not in the pull-down, confirming the specificity of the CKAP5 labeling. The new data is presented in Fig. 3G and Supp. Fig. 3F.

Regarding the difference in the γ -tubulin signal, this outcome is reproducible and also observed in the new experiment. Biotinylation of γ -tubulin (and consequently the efficiency of the pull-down) depends on the accessibility of lysine residues on γ -tubulin's surface and the relative positioning of the BirA biotin ligase. Considering that two different bait proteins (GCP2 vs. GCP3) and different types of BirA fusions (N- vs. C-terminal) were used, we do not consider this outcome unusual. The important result is that in both cases γ -tubulin is specifically labeled and pulled-down.

Minor points:

6) Why was the role of the known CKAP5/ch-TOG partner, TACC3 not investigated? TACC3 is known to interact with the C-terminus of CKAP5/ch-TOG and is known to be important for its different functions at the centrosomes and in the mitotic spindle (Thakur et al., J Biol Chem, 2014). TACC3 could be an illuminating target for further study.

We agree that TACC3 would be an interesting factor to analyze. However, it has been characterized mostly in the context of mitotic spindle assembly, k-fiber stability, and microtubule plus-end regulation. Characterizing TACC3 in the context of interphase centrosomes and the Golgi would require a significant amount of additional work that is best suited, as suggested, for a future study.

7) Line 179: "strong overall decrease"; Figure 3B indicates this is a 50% decrease. Please use more specific language (e.g. 50% decrease) instead of "strong overall decrease."

8) Line 188: "strongly reduced"; same issue as point 7.

9) Lines 221-222: "Some microtubules..."; same issue as point 7.

10) Lines 230-231: "...some regrown microtubules coincided with NIN..."; same issue as point 7.

We have modified the indicated text passages by using more precise language.

Reviewer #3:

In this paper, the authors examine the localisation of the human TOG domain protein, CKAP5, and the effect of its depletion on γ -TuRC localisation and microtubule nucleation at interphase centrosomes in cultured cells. They find that CKAP5 localises predominantly to sub-distal appendages, but also localises transiently to the interphase PCM – localisation to the interphase PCM is observed only after microtubule depolymerisation and depends upon the c-terminal (non-TOG domain) region – consistent with this, they identify a transient interaction between CKAP5 and γ -TuRC via a bio-ID assay, but do not observe a stable interaction via an IP assay. They also show that depletion of CKAP5 leads to a reduction in γ -TuRC localisation to the interphase PCM and to reduced nucleation. Most studies of TOG domain proteins have either been performed in vitro or on CKAP5 homologues and so it is refreshing to see a study on human CKAP5 (also known as ch-TOG) within cells. Nevertheless, much of what has been observed in this study is already known for TOG proteins in general i.e. they can interact with γ -TuRC and promote microtubule nucleation. New insight, however, comes from the observation that CKAP5 is required for γ -TuRC localisation to the interphase PCM, which is an unexpected result. It remains unclear why this is the case and whether it is specific to the interphase PCM. Indeed, the authors mention that a previous paper showed there was no effect on γ -tubulin levels at mitotic centrosomes after CKAP5 depletion. It would have been nice if the authors had confirmed this here or shown that there is in fact a quantifiable effect at mitotic centrosomes – it seems like a very doable experiment. Overall, the results are solid and presented clearly but we do not learn a great deal about how CKAP5 functions. It should be an editorial decision as to whether they feel the paper is suitable for publication in Nature Communications. I have a few comments listed below that the authors should address.

We thank the reviewer for this fair assessment. As also outlined in more detail below, we note that we have added new data to the manuscript that expands its scope, including an analysis of CKAP5's role in γ TuRC recruitment at mitotic centrosomes (Supp. Fig. 5) and at the Golgi (Fig. 7).

1) I think it is important that the text reflects the fact that the authors have studied CKAP5 only at interphase centrosomes. There are examples of differences between interphase and mitotic centrosomes in the literature, and the centrosome is only one of many different MTOCs. The current title is “CKAP5-dependent localization and activity of the microtubule nucleator γ TuRC”. I don't think the authors provide any data regarding the activity of γ -TuRCs, but rather show that CKAP5 is required for localisation of γ -TuRC to interphase centrosomes. A more apt title would be “CKAP5-dependent localization of the microtubule nucleator γ TuRC at interphase centrosomes”.

We have changed the title to more accurately describe the main findings. The new title is: “Microtubule nucleation and γ TuRC centrosome localization in interphase cells require CKAP5”.

In the abstract the authors say: “Here we show that at centrosomes, rather than adapters, the microtubule polymerase CKAP5 ultimately controls gTuRC recruitment and activation”. They should say “Here we show that at interphase centrosomes, rather than adapters, the microtubule polymerase CKAP5 ultimately controls gTuRC recruitment and activation”. Also “In the absence of CKAP5, gTuRC fails to localize to centrosomal nucleation sites” should read “In the absence of CKAP5, gTuRC fails to localize to centrosomal nucleation sites during interphase”. And “Together our data show that centrosome attachment of gTuRC and activation of nucleation are mechanistically coupled through transient recruitment of CKAP5” should be “Together our data show that attachment of gTuRC at interphase centrosomes are mechanistically coupled through transient recruitment of CKAP5”. In Discussion the authors say: “Recruitment and activation of the microtubule nucleator gTuRC are fundamental to the function of MTOCs. In this study we show that at centrosomes in human cells the XMAP215 family member CKAP5 is crucial for both of these functions”, when this should be “In this study we show that at interphase centrosomes in human cells the XMAP215 family member CKAP5 is crucial for γ -TuRC recruitment”. Also in the Discussion the authors say: “In contrast, our results indicate that in human cells CKAP5 is crucial for stable MTOC incorporation of gTuRC through a mechanism that involves stimulation of nucleation”. The authors cannot generalise to all MTOCs. Then they say “This is further supported by our finding that CKAP5 centrosome localization, in turn, also depends on gTuRC” – needs to be ...centrosome localization to interphase centrosomes.

The reviewer is correct regarding the focus of the manuscript on interphase and that this was not always made clear in the text. In the revised manuscript we have added new data on CKAP5's role in γ TuRC recruitment at mitotic centrosomes (Supp. Fig. 5) and in nucleation from the Golgi (Fig. 7). We now refer to ‘interphase centrosomes’ in several sections including in the abstract, whenever the data are restricted to this organelle and cell cycle stage.

2) Supp Fig 1A – the authors say that “Only in late G2 phase CKAP5 was also detected at more proximal centriole regions, in addition to its localization to subdistal appendages (Supp. Fig. 1A)” but I find it difficult to see a difference in the localisation pattern through the cell cycle from the images shown. Perhaps the authors can point these differences out more clearly.

We have modified Supp. Fig. 1A and included arrowheads to point at the differences between CKAP5 labeling at different cell cycle stages. To improve clarity, we have also removed early S phase examples and now only compare the two extremes, G1 vs. S/G2. In the presence of microtubules, in G1 cells, CKAP5 staining is restricted to the subdistal appendage region of only one of the two centrioles. In contrast, in S/G2 cells CKAP5 is not restricted to this region and clearly present on both centrioles (now each carrying an engaged daughter centriole). At both cell cycle stages the CKAP5 signal accumulates around centrioles upon depolymerization of microtubules. This accumulation is also observed with the second CKAP5 antibody, for which we have also added an example in Supp. Fig. 1A.

3) It would be useful for the authors to quantify CKAP5 levels at centrosomes after CKAP5 RNAi.

We have performed this quantification and included it in Fig. 1C.

4) How were intensity measurements made exactly? Are they mean intensity measurements or sum intensity measurements?

We have added this information to the methods.

5) The authors say that CKAP5 and γ -tubulin colocalised: “CKAP5 accumulation occurred mostly in the proximal centriole region, where it colocalized with γ -tubulin suggesting that it was present in the PCM (Fig. 2A, white arrows; Supp. Fig. 2B)”, but it seems that the signals do not perfectly overlap. This could probably best be seen with line scans. I think the authors should check this carefully and re-phase the sentence if necessary.

Indeed, while they overlap significantly, CKAP5 and γ -tubulin do not perfectly colocalize. We have rephrased our description in the corresponding passages to acknowledge this fact.

6) The experiment in Fig 6C is not cited in the main text. Moreover, it is unclear where the non-centrosomal microtubules are originating from – are these from the Golgi, as the authors mentioned that these cells nucleate some microtubules from the Golgi? Or are they forming randomly in the cytoplasm? If the authors can determine this, they can then expand their analysis to non-centrosomal sites e.g. they could assess microtubule nucleation from these sites after CKAP5 knockdown. This would expand the breadth of the paper outside of just interphase centrosomes.

This is an excellent suggestion and we have performed additional experiments to address this point in the revised manuscript. In RPE1 cells the majority of non-centrosomal, cytoplasmic microtubules during regrowth indeed originate from the Golgi. Our new data show (i) that CKAP5 is observed in association with short microtubules at Golgi membranes during the nucleation phase and (ii) that CKAP5 depletion reduces Golgi-associated microtubule nucleation. As indicated by the reviewer, these data expand the breadth of the study, implicating CKAP5 not only in γ TuRC-dependent nucleation from interphase centrosomes but also from the Golgi.

References

- Consolati, T., Locke, J., Roostalu, J., Chen, Z. A., Gannon, J., Asthana, J., Lim, W. M., Martino, F., Cvetkovic, M. A., Rappsilber, J., et al. (2020). Microtubule Nucleation Properties of Single Human γ TuRCs Explained by Their Cryo-EM Structure. *Dev Cell* 53, 603-617.e8.
- Ezquerra, A., Viais, R. and Lüders, J. (2020). Assaying Microtubule Nucleation. *Methods Mol Biol* 2101, 163–178.
- Fong, K.-W., Choi, Y.-K., Rattner, J. B. and Qi, R. Z. (2008). CDK5RAP2 is a pericentriolar protein that functions in centrosomal attachment of the gamma-tubulin ring complex. *Mol Biol Cell* 19, 115–125.

- Gavilan, M. P., Gandolfo, P., Balestra, F. R., Arias, F., Bornens, M. and Rios, R. M. (2018). The dual role of the centrosome in organizing the microtubule network in interphase. *EMBO Rep* 19, e45942.
- Kollman, J. M., Greenberg, C. H., Li, S., Moritz, M., Zelter, A., Fong, K. K., Fernandez, J.-J., Sali, A., Kilmartin, J., Davis, T. N., et al. (2015). Ring closure activates yeast γ TuRC for species-specific microtubule nucleation. *Nat Struct Mol Biol* 22, 132–137.
- Liu, P., Zupa, E., Neuner, A., Böhler, A., Loerke, J., Flemming, D., Ruppert, T., Rudack, T., Peter, C., Spahn, C., et al. (2020). Insights into the assembly and activation of the microtubule nucleator γ -TuRC. *Nature* 578, 467–471.
- Schweizer, N., Haren, L., Dutto, I., Viais, R., Lacasa, C., Merdes, A. and Lüders, J. (2021). Sub-centrosomal mapping identifies augmin- γ TuRC as part of a centriole-stabilizing scaffold. *Nat Commun* 12, 6042.
- Tovey, C. A., Tsuji, C., Egerton, A., Bernard, F., Guichet, A., de la Roche, M. and Conduit, P. T. (2021). Autoinhibition of Cnn binding to γ -TuRCs prevents ectopic microtubule nucleation and cell division defects. *J Cell Biol* 220, e202010020.
- Wieczorek, M., Urnavicius, L., Ti, S.-C., Molloy, K. R., Chait, B. T. and Kapoor, T. M. (2020). Asymmetric Molecular Architecture of the Human γ -Tubulin Ring Complex. *Cell* 180, 165-175.e16.
- Wu, J., de Heus, C., Liu, Q., Bouchet, B. P., Noordstra, I., Jiang, K., Hua, S., Martin, M., Yang, C., Grigoriev, I., et al. (2016). Molecular Pathway of Microtubule Organization at the Golgi Apparatus. *Dev Cell* 39, 44–60.
- Zimmermann, F., Serna, M., Ezquerra, A., Fernandez-Leiro, R., Llorca, O. and Luders, J. (2020). Assembly of the asymmetric human γ -tubulin ring complex by RUVBL1-RUVBL2 AAA ATPase. *Sci Adv* 6, eabe0894.

REVIEWER COMMENTS

Reviewer #1 (Remarks to the Author):

In Microtubule nucleation and γ TuRC centrosome localization in interphase cells require CKAP5 Ali et al. present extensive imaging and molecular evidence about the role of CKAP5 in microtubule formation, particularly around the centriole. Overall this is a good paper presents a clear story with extensive evidence. They have covered the reviewers comments well and the paper has improved significantly.

Again I wish to praise the authors for their clear statistics, ensuring every claim has repeats, N numbers, statistical tests and clear descriptions of plots such as box and whisker plots. They also include complete original images of gels to clearly demonstrate the truth of their claims.

Remaining major issues:

The authors use maximum intensity projections of the 3D data to calculate intensity values. This is extremely bad practice, as the maximum intensity projection preferentially selects large positive noise values, this particularly enhances differences between signal and background when the major noise factor is shot noise from the photon statistics. Maximum intensity projections and a very useful tool for visualizing the the complex 3D data as a 2D summary, and this enhancement of signal over background helps visualize these possible small differences, however these images should never be used for numeric analysis as they non-linearly enhance the differences. The numeric analysis should be done in 3D or using a mean or summed projection to avoid this issue.

"super resolution confocal microscope Elyra PS1 (LSM 880) that achieves 100nm XY and 200nm Z-axis optical resolutions." The Elyra is not a confocal microscope. If the authors wish to claim an achieved resolution they need to measure it, the claim of 100nm in XY and 200nm in Z is straight from the manufactures publicity documents and not realistic.

Minor issues:

The paper still does not appear to say at any point what protein is labeled at which wavelength for each set of experiments. Maybe this is represented by the display colors, but this may not be true and doesn't appear to be stated anywhere. This is a significant issue as the longer wavelength channels have resolutions 30-50% worse than the shorter wavelength channels, the SIM super resolution reduces the resolution but it still scales linearly with wavelength.

Reviewer #2 (Remarks to the Author):

The authors sufficiently addressed most points raised. There remain the following points of discussion:

Major comments:

1) I am still not convinced that the use of CKAP5 as the main protein name provides a large enough benefit over the far more commonly used designation of ch-TOG. A cursory search on PubMed shows the use of ch-TOG in 64 studies, 46 of which have been published in the last ten years (including 2022). In contrast, CKAP5 is not used in the field. There is no point in adding confusion by switching to another protein name now.

2) I appreciate the authors' CDK5RAP2 KO cell-based experiments and agree that, in interphase centrosomes, the contribution of this adapter to gTuRC recruitment does not seem to be significant for the cell lines tested.

3) In Figure 7A, the authors state that after nocodazole treatment and MT re-growth of RPE1 cells, "CKAP5 signals showed some enrichment at short microtubules in [Golgi] clusters, consistent with specific recruitment..." (lines 308-309). They also note that they couldn't assign these CKAP5/ch-TOG signals to specific MT ends, which is understandable. Presumably the purpose of Figure 7A was to show that CKAP5/ch-TOG co-localizes with Golgi membranes and MTs, and thus bolster the idea that it might play a role in promoting Golgi-mediated MT nucleation. However, upon closer inspection of the image presented in Fig. 7A, I do not agree that there is significant association of CKAP5/ch-TOG signal with the Golgi clusters. The CKAP5/ch-TOG signals appear to be present throughout the majority of the image (with the predominant nuclear stain mentioned by the authors). Similarly, there appear to be short MTs everywhere, whether there is a visible Golgi membrane cluster nearby or not. In all, the image doesn't appear particularly informative for the reader. I would suggest moving this to a supplemental figure, and instead highlight the more convincing data represented in Fig 7B and 7C which does show a role for CKAP5/ch-TOG in Golgi MT nucleation.

4) The authors should include some quantification to justify the statement made in lines 308-309 that "CKAP5 signals showed some enrichment at short microtubules...consistent with specific recruitment." The signal doesn't seem to be specifically recruited by eye, and quantification could help convince the reader. Also, using more specific language than "some enrichment" would be preferable.

Reviewer #3 (Remarks to the Author):

I think the authors have done an excellent job in revising the paper. It is ready for publication.

Response to the reviewers' comments

Reviewer #1 (Remarks to the Author):

In Microtubule nucleation and γ TuRC centrosome localization in interphase cells require CKAP5 Ali et al. present extensive imaging and molecular evidence about the role of CKAP5 in microtubule formation, particularly around the centriole. Overall this is a good paper presents a clear story with extensive evidence. They have covered the reviewers comments well and the paper has improved significantly.

Again I wish to praise the authors for their clear statistics, ensuring every claim has repeats, N numbers, statistical tests and clear descriptions of plots such as box and whisker plots. They also include complete original images of gels to clearly demonstrate the truth of their claims.

Remaining major issues:

The authors use maximum intensity projections of the 3D data to calculate intensity values. This is extremely bad practice, as the maximum intensity projection preferentially selects large positive noise values, this particularly enhances differences between signal and background when the major noise factor is shot noise from the photon statistics. Maximum intensity projections and a very useful tool for visualizing the the complex 3D data as a 2D summary, and this enhancement of signal over background helps visualize these possible small differences, however these images should never be used for numeric analysis as they non-linearly enhance the differences. The numeric analysis should be done in 3D or using a mean or summed projection to avoid this issue.

In our experience max projection quantifications work reasonably well for very small structures such as centrioles/centrosomes. However, we agree that this approach is not ideal and we have re-quantified intensities in all experiments with sum projections. While the absolute values are slightly different compared to our previous quantifications, there are no major changes in the relative differences between samples and the drawn conclusions.

"super resolution confocal microscope Elyra PS1 (LSM 880) that achieves 100nm XY and 200nm Z-axis optical resolutions." The Elyra is not a confocal microscope. If the authors wish to claim an achieved resolution they need to measure it, the claim of 100nm in XY and 200nm in Z is straight from the manufactures publicity documents and not realistic.

We agree that the statement about resolution has no practical value for our analyses and we have removed it. Instead we now state:

“Three dimensional-structured illumination microscopy (3D-SIM) was performed on a super resolution microscope Elyra PS.1 (Carl Zeiss, Germany).”

Minor issues:

The paper still does not appear to say at any point what protein is labeled at which wavelength for each set of experiments. Maybe this is represented by the display colors, but this may not be true and doesn't appear to be stated anywhere. This is a significant issue as the longer wavelength channels have resolutions 30-50% worse than the shorter wavelength channels, the SIM super resolution reduces the resolution but it still scales linearly with wavelength.

We have added a statement to the methods that clarifies this. We would like to emphasize that that the achieved resolutions were always sufficient to support our conclusions.

Reviewer #2 (Remarks to the Author):

The authors sufficiently addressed most points raised. There remain the following points of discussion:

Major comments:

1) I am still not convinced that the use of CKAP5 as the main protein name provides a large enough benefit over the far more commonly used designation of ch-TOG. A cursory search on PubMed shows the use of ch-TOG in 64 studies, 46 of which have been published in the last ten years (including 2022). In contrast, CKAP5 is not used in the field. There is no point in adding confusion by switching to another protein name now.

The reviewer's suggested term “ch-TOG” is equally frequent with “chTOG” (without hyphen) (66 entries) and the results include non-overlapping entries. Compared to these terms, searching PubMed with “CKAP5” finds 147 entries. That being said, we agree that this term is less common in the centrosome/spindle/microtubule fields. We now use “ch-TOG”, which is the originally used spelling, throughout the paper including figures. In addition, we mention “chTOG” and “CKAP5” as alternative names.

2) I appreciate the authors' CDK5RAP2 KO cell-based experiments and agree that, in interphase centrosomes, the contribution of this adapter to gTuRC recruitment does not seem to be significant for the cell lines tested.

3) In Figure 7A, the authors state that after nocodazole treatment and MT re-growth of RPE1 cells, “CKAP5 signals showed some enrichment at short microtubules in [Golgi] clusters, consistent with specific recruitment...” (lines 308-309). They also note that they couldn't assign these CKAP5/ch-TOG signals to specific MT ends, which is understandable. Presumably the purpose of Figure 7A was to show that CKAP5/ch-TOG co-localizes with Golgi membranes and MTs, and thus bolster the idea that it might play a role in

promoting Golgi-mediated MT nucleation. However, upon closer inspection of the image presented in Fig. 7A, I do not agree that there is significant association of CKAP5/ch-TOG signal with the Golgi clusters. The CKAP5/ch-TOG signals appear to be present throughout the majority of the image (with the predominant nuclear stain mentioned by the authors). Similarly, there appear to be short MTs everywhere, whether there is a visible Golgi membrane cluster nearby or not.

In all, the image doesn't appear particularly informative for the reader. I would suggest moving this to a supplemental figure, and instead highlight the more convincing data represented in Fig 7B and 7C which does show a role for CKAP5/ch-TOG in Golgi MT nucleation.

We have repeated the Golgi regrowth experiments with different timings and fixation conditions. However, we could not improve the visualization of ch-TOG co-localization with microtubules at Golgi clusters to allow reliable quantification. Limitations are the available antibodies, cytoplasmic background staining, and the presumably transient nature of the colocalization. We have followed the reviewer's suggestion and moved the panels to the supplementary data, instead emphasizing the measured effect on Golgi microtubule nucleation.

4) The authors should include some quantification to justify the statement made in lines 308-309 that "CKAP5 signals showed some enrichment at short microtubules...consistent with specific recruitment." The signal doesn't seem to be specifically recruited by eye, and quantification could help convince the reader. Also, using more specific language than "some enrichment" would be preferable.

We have moved this analysis to the supplementary data (see point 3 above) and have removed "enrichment" and "specific recruitment" from this statement, since we cannot support it with quantitative data.

Reviewer #3 (Remarks to the Author):

I think the authors have done an excellent job in revising the paper. It is ready for publication.

REVIEWERS' COMMENTS

Reviewer #1 (Remarks to the Author):

The authors have addressed my issues with previous versions of the manuscript and I think it is now suitable for publication